# WHITENING AND SECOND ORDER OPTIMIZATION BOTH DESTROY INFORMATION ABOUT THE DATASET, AND CAN MAKE GENERALIZATION IMPOSSIBLE

## ABSTRACT

Machine learning is predicated on the concept of generalization: a model achieving low error on a sufficiently large training set should also perform well on novel samples from the same distribution. We show that both data whitening and second order optimization can harm or entirely prevent generalization. In general, model training harnesses information contained in the sample-sample second moment matrix of a dataset. For a general class of models, namely models with a fully connected first layer, we prove that the information contained in this matrix is the *only information* which can be used to generalize. Models trained using whitened data, or with certain second order optimization schemes, have less access to this information; in the high dimensional regime they have no access at all, resulting in poor or nonexistent generalization ability. We experimentally verify these predictions for several architectures, and further demonstrate that generalization continues to be harmed even when theoretical requirements are relaxed. However, we also show experimentally that *regularized* second order optimization can provide a practical tradeoff, where training is accelerated but less information is lost, and generalization can in some circumstances even improve.

## 1 INTRODUCTION

Whitening is a data preprocessing step that removes correlations between input features (see Fig. 1). It is used across many scientific disciplines, including geology (Gillespie et al., 1986), physics (Jenet et al., 2005), machine learning (Le Cun et al., 1998), linguistics (Abney, 2007), and chemistry (Bro & Smilde, 2014). It has a particularly rich history in neuroscience, where it has been proposed as a mechanism by which biological vision realizes Barlow's redundancy reduction hypothesis (Attneave, 1954; Barlow, 1961; Atick & Redlich, 1992; Dan et al., 1996; Simoncelli & Olshausen, 2001).

Whitening is often recommended since, by standardizing the variances in each direction in feature space, it typically speeds up the convergence of learning algorithms (Le Cun et al., 1998; Wiesler & Ney, 2011), and causes models to better capture contributions from low variance feature directions. Whitening can also encourage models to focus on more fundamental higher order statistics in data, by removing second order statistics (Hyvärinen et al., 2009). Whitening has further been a direct inspiration for deep learning techniques such as batch normalization (Ioffe & Szegedy, 2015) and dynamical isometry (Pennington et al., 2017; Xiao et al., 2018).

### 1.1 WHITENING DESTROYS INFORMATION USEFUL FOR GENERALIZATION

In the high dimensional setting, for any model with a fully connected first layer, we show theoretically and experimentally that whitening the data and then training with gradient descent or stochastic gradient descent (SGD) results in a model with poor or nonexistent generalization ability, depending on how the whitening transform is computed. We emphasize that, analytically, this result applies to any model whose first layer is fully connected, and is not restricted to linear models. Empirically, the results hold in an even larger context, including in convolutional networks. Here, the high dimensional setting corresponds to a number of input features which is comparable to or larger than the number of datapoints. While this setting does not usually arise in modern neural network applications, it is of particular relevance in fields where data collection is expensive or otherwise prohibitive (Levesque

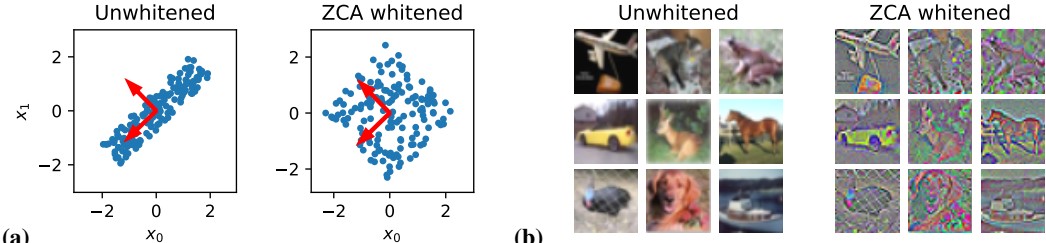

Figure 1: **Whitening removes correlations between feature dimensions in a dataset.** Whitening is a linear transformation of a dataset that sets all non-zero eigenvalues of the covariance matrix to 1. ZCA whitening is a specific choice of the linear transformation that rescales the data in the directions given by the eigenvectors of the covariance matrix, but without additional rotations or flips. *(a)* A toy 2d dataset before and after ZCA whitening. Red arrows indicate the eigenvectors of the covariance matrix of the unwhitened data. *(b)* ZCA whitening of CIFAR10 images preserves spatial and chromatic structure, while equalizing the variance across all feature directions.

et al., 2012), or where the data is intrinsically high dimensional (Stringer et al., 2019; Fusi et al., 2016; Shyr, 2012; Martínez-Ramón et al., 2006; Bruce et al., 2002), and is also the focus of increasing interest in statistics (Wainwright, 2019).

The loss of generalization ability for high dimensional whitened datasets is due to the fact that whitening destroys information in the dataset, and that *in high dimensional datasets whitening destroys all information which can be used for prediction.* This is related to investigations of information loss due to PCA projection (Geiger & Kubin, 2012). Our result is not restricted to neural networks, and applies to any model in which the input is transformed by a dense matrix with isotropic weight initialization.

### 1.2 SECOND ORDER OPTIMIZATION HARMS GENERALIZATION SIMILARLY TO WHITENING

Second order optimization algorithms take advantage of information about the curvature of the loss landscape to take a more direct route to a minimum (Boyd & Vandenberghe, 2004; Bottou et al., 2018). There are many approaches to second order or quasi-second order optimization (Martens & Grosse, 2015; Dennis Jr & Moré, 1977; Broyden, 1970; Fletcher, 1970; Goldfarb, 1970; Shanno, 1970; Liu & Nocedal, 1989; Schraudolph et al., 2007; Sunehag et al., 2009; Martens, 2010; Byrd et al., 2011; Vinyals & Povey, 2011; Lin et al., 2008; Hennig, 2013; Byrd et al., 2014; Sohl-Dickstein et al., 2014; Desjardins et al., 2015; Grosse & Martens, 2016; Martens et al., 2018; George et al., 2018; Zhang et al., 2017; Botev et al., 2017; Bollapragada et al., 2018; Berahas et al., 2019; Gupta et al., 2018; Agarwal et al., 2016; Duchi et al., 2011; Shazeer & Stern, 2018; Anil et al., 2019; Agarwal et al., 2019; Lu et al., 2018; Kingma & Ba, 2014; Zeiler, 2012; Tieleman & Hinton, 2012; Osawa et al., 2020), and there is active debate over whether second order optimization harms generalization (Wilson et al., 2017; Zhang et al., 2018; 2019; Amari et al., 2020; Vaswani et al., 2020). The measure of curvature used in these algorithms is often related to feature-feature covariance matrices of the input, and of intermediate activations (Martens & Grosse, 2015). In some situations, it is already known that second order optimization is equivalent to steepest descent training on whitened data (Sohl-Dickstein, 2012; Martens & Grosse, 2015).

The similarities between whitening and second order optimization allow us to argue that pure second order optimization also prevents information about the input distribution from being leveraged during training, and can harm generalization (see Figs. 3, 4). We do find, however, that when strongly regularized and carefully tuned, second order methods can lead to superior performance (Fig. 5).

## 2 THEORY OF WHITENING, SECOND ORDER OPTIMIZATION, AND GENERALIZATION

Consider a dataset $X \in \mathbb{R}^{d \times n}$ consisting of $n$ independent $d$-dimensional examples. We write $F$ for the feature-feature second moment matrix and $K$ for the sample-sample second moment matrix:

$$F = XX^\top \in \mathbb{R}^{d \times d}, \quad K = X^\top X \in \mathbb{R}^{n \times n}. \tag{1}$$

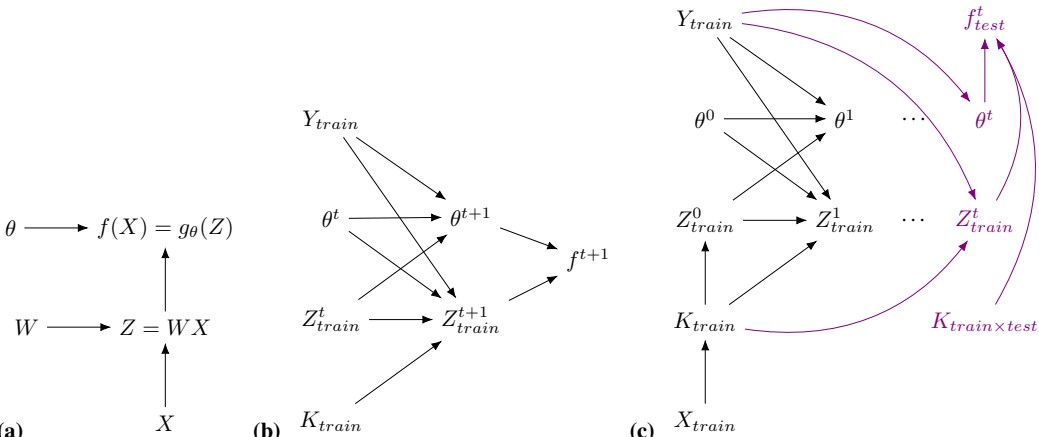

Figure 2: **Activations and weights depend on the training data only through second moments.** *(a)* Our model class consists of a linear transformation $Z = WX$, followed by a nonlinear map $g_\theta(Z)$ with parameters $\theta$. Note that this model class includes fully connected neural networks, among other common machine learning models. *(b)* Causal dependencies for a single gradient descent update. The changes in weights, activations, and model output depend on the training data through the training sample second moment matrix, $K_{\text{train}}$, and the targets, $Y_{\text{train}}$. *(c)* Causal structure for the entire training trajectory. The final weights and training activations only depend on the training data through the training sample second moment matrix $K_{\text{train}}$, and the targets $Y_{\text{train}}$, while the test predictions (in purple) also depend on the mixed second moment matrix, $K_{\text{train}\times\text{test}}$.

We assume that at least one of $F$ or $K$ is full rank. We omit normalization factors of $1/n$ and $1/d$ in the definitions of $F$ and $K$, respectively, for notational simplicity in later sections. Note that as defined, $K$ is also the Gram matrix of $X$.

**Definition 2.0.1** (Whitening). *Any linear transformation $M$ s.t. $\hat{X} = MX$ maps the eigenspectrum of $F$ to ones and zeros, with the multiplicity of ones given by* $\text{rank}(F)$.

We consider the two cases $n \leq d$ and $n \geq d$ (when $n = d$ both cases apply).

$$n \geq d: \ \hat{F} = I^{d\times d}, \quad \hat{K} = \sum_{i=1}^{d} \hat{u}_i \hat{u}_i^\top.$$

$$n \leq d: \ \hat{F} = \sum_{j=1}^{n} \hat{v}_j \hat{v}_j^\top, \quad \hat{K} = I^{n\times n}. \tag{2}$$

Here, $\hat{F}$ and $\hat{K}$ denote the whitened second moment matrices, and the vectors $\hat{u}_i$ and $\hat{v}_j$ are orthogonal unit vectors of dimension $n$ and $d$ respectively. Eq. 2 follows directly from the fact that $X^\top X$ and $XX^\top$ share nonzero eigenvalues.

We are interested in understanding the effect of whitening on the performance of a trained model when evaluated on a test set. We will see that for models with a dense first layer (eg, fully connected neural networks), the trained model depends on the training inputs only through $K$. In general, training dynamics and generalization performance can depend non-trivially on $K$. However, whitening trivializes $K$, and so eliminates the ability of the network and training algorithm to take advantage of information contained in it.

## 2.1 TRAINING DYNAMICS DEPEND ON THE TRAINING DATA ONLY THROUGH ITS SECOND MOMENTS

Consider a model $f$ with a dense first layer $Z$:

$$f(X) = g_\theta(Z), \quad Z = WX, \tag{3}$$

where $W$ denotes the first layer weights and $\theta$ denotes all remaining parameters (see Fig. 2(a)). The structure of $g_\theta(\cdot)$ is unrestricted. $W$ is initialized from an isotropic distribution. We study a

supervised learning problem, in which each vector $X_i$ corresponds to a label $Y_i$.[1] We adopt the notation $X_{\text{train}} \in \mathbb{R}^{d \times n_{\text{train}}}$ and $Y_{\text{train}}$ for the training inputs and labels, and write the corresponding second moment matrices as $F_{\text{train}}$ and $K_{\text{train}}$. We consider models with loss $L(f(X); Y)$ trained by SGD. The update rules are

$$\theta^{t+1} = \theta^t - \eta \frac{\partial L^t}{\partial \theta^t} \ \text{ and } \ W^{t+1} = W^t - \eta \frac{\partial L^t}{\partial W^t} = W^t - \eta \frac{\partial L^t}{\partial Z_{\text{train}}^t} X_{\text{train}}^\top , \tag{4}$$

where $t$ denotes the current training step, $\eta$ is the learning rate, and $L^t$ is the loss evaluated only on the minibatch used at step $t$.

As a result, the activations $Z_{\text{train}}$ evolve as

$$Z_{\text{train}}^{t+1} = Z_{\text{train}}^t - \eta \frac{\partial L^t}{\partial Z_{\text{train}}^t} X_{\text{train}}^\top X_{\text{train}} = Z_{\text{train}}^t - \eta \frac{\partial L^t}{\partial Z_{\text{train}}^t} K_{\text{train}}. \tag{5}$$

Treating the weights, activations, and function predictions as random variables, with distributions induced by the initial distribution over $W^0$, the update rules (Eqs. 4-5) can be represented by the causal diagram in Fig. 2(b). We can now state one of our main results.

**Theorem 2.1.1.** *Let $f(X)$ be a function as in Eq. 3, with linear first layer $Z = WX$, and additional parameters $\theta$. Let $W$ be initialized from an isotropic distribution. Further, let $f(X)$ be trained via gradient descent on a training dataset $X_{train}$. The learned weights $\theta^t$ and first layer activations $Z_{train}^t$ are independent of $X_{train}$ conditioned on $K_{train}$ and $Y_{train}$. In terms of mutual information $\mathcal{I}$, we have*

$$\mathcal{I}(Z_{train}^t, \theta^t; X_{train} \mid K_{train}, Y_{train}) = 0 \ \forall t. \tag{6}$$

*Proof.* To establish this result, we note that the first layer activation at initialization, $Z_{\text{train}}^0$, is a random variable due to random weight initialization, and only depends on $X_{\text{train}}$ through $K_{\text{train}}$:

$$\mathcal{I}(Z_{\text{train}}^0; X_{\text{train}} \mid K_{\text{train}}) = 0. \tag{7}$$

This is a consequence of the isotropy of the initial weight distribution, explained in detail in Appendix A. Combining this with Eqs. 4-5, the causal diagram for all of training is given by (the black part of) Fig. 2(c). The conditional independence of Eq. 6 follows from this diagram. $\qquad \square$

## 2.2 Test set predictions depend on train and test inputs only through their second moments

Let $X_{\text{test}} \in \mathbb{R}^{d \times n_{\text{test}}}$ and $Y_{\text{test}}$ be the test data. The test predictions $f_{\text{test}} = f(X_{\text{test}})$ are determined by $Z_{\text{test}}^t = W^t X_{\text{test}}$ and $\theta^t$. So far we have discussed the evolution of $Z_{\text{train}}$. To identify sources of data dependence, we can write the evolution of the test set predictions $Z_{\text{test}}$ over the course of training in a similar fashion:

$$Z_{\text{test}}^{t+1} = Z_{\text{test}}^t - \eta \frac{\partial L^t}{\partial Z_{\text{train}}^t} K_{\text{train} \times \text{test}}, \tag{8}$$

where $K_{\text{train} \times \text{test}} = X_{\text{train}}^\top X_{\text{test}} \in \mathbb{R}^{n_{\text{train}} \times n_{\text{test}}}$. The initial first layer activations are independent of the training data, and depend on $X_{\text{test}}$ only through $K_{\text{test}}$:

$$\mathcal{I}(Z_{\text{test}}^0; X \mid K_{\text{test}}) = 0, \tag{9}$$

where $X$ is the combined training and test data. If we denote the second moment matrix over this combined set by $K$, then the evolution of the test predictions is described by the (purple part of the) causal diagram in Fig. 2(c), from which we conclude the following.

**Theorem 2.2.1.** *For a function $f(X)$ as in Eq. 3, trained with the update rules Eqs. 4-5 from an isotropic weight initialization, test predictions depend on the training data only through $K$ and $Y_{train}$. This is summarized in the mutual information statement*

$$\mathcal{I}(f_{test}; X \mid K, Y_{train}) = 0. \tag{10}$$

We emphasize that Theorem 2.2.1 applies to any model with a dense first layer, and is not limited to linear models.

---

[1] Our results also apply to unsupervised learning, which can be viewed as a special case of supervised learning where $Y_i$ contains no information.

### 2.3 Whitening harms generalization in high dimensional datasets

**Full data whitening of a high dimensional dataset.** We first consider a simplified setup: computing the whitening transform using the combined training and test data. We refer to this as 'full-whitening'. We consider the large feature count ($d \geq n$) regime. In this case, by Eq. 2 we have $K = I$. Since $K$ is now a constant rather than a stochastic variable, Eq. 10 reduces to

$$\mathcal{I}(f_{\text{test}}; \hat{X} \mid Y_{\text{train}}) = 0. \tag{11}$$

That is, *test set predictions contain no information about the model inputs $X$*. Model accuracy in this regime can be no higher than chance.

**Training data whitening of a high dimensional dataset.** In practice, we are more interested in the common setting of computing a whitening transform based only on the training data. We call data whitened in this way 'train-whitened'. As mentioned above, the test predictions of a model are entirely determined by the first layer activations $Z_{\text{test}}^t$ and the weights $\theta^t$. From Theorem 2.1.1 we see that the learned weights $\theta^t$ depend on the training data only through $K_{\text{train}}$, and are thus independent of the training data for whitened data:

$$\mathcal{I}(\theta^t; \hat{X}_{\text{train}} \mid Y_{\text{train}}) = 0. \tag{12}$$

It is worth emphasizing this point because in most realistic networks the majority of model parameters are contained in these deeper weights $\theta^t$.

Despite the deep layer weights, $\theta^t$, being unable to extract information from the training distribution, the model is not entirely incapable of generalizing to test inputs. This is because the test activations $Z_{\text{test}}$ will interpolate between training examples, using the information in $\hat{K}_{\text{train} \times \text{test}}$. More precisely,

$$Z_{\text{test}}^t = Z_{\text{test}}^0 + \left( Z_{\text{train}}^t - Z_{\text{train}}^0 \right) \hat{K}_{\text{train} \times \text{test}}. \tag{13}$$

This interpolation in $Z$ is the *only* way in which structure in the inputs $X_{\text{train}}$ can drive generalization. This should be contrasted with the case of full data whitening, discussed above, where $\hat{K}_{\text{train} \times \text{test}} = 0$. We therefore predict that when whitening is performed only on the training data, there will be some generalization, but it will be much more limited than can be achieved without whitening.

### 2.4 Whitening in linear least squares models

Linear models $f = WX$ provide intuition on why whitening can be harmful, which we discuss briefly here. The linear case also enables us to extend our theory to also apply to low dimensional datasets. A detailed exposition is in Appendix B.

For this section only, consider the low dimensional case $d < n$, where the loss has a unique global optimum $W^\star$. The model predictions at this optimum are invariant to whitening. However, whitening has an effect on the dynamics of model predictions over the course of training. When training is performed with early stopping based on validation loss, predictions differ considerably for models trained on whitened and unwhitened data. These benefits from early stopping can be related to benefits from weight regularization (Yao et al., 2007).

We focus on the continuous-time picture because it is the clearest, but similar statements can be made for gradient descent. Recall that $v_i$ are the eigenvectors of $F_{\text{train}}$. Denoting the corresponding eigenvalues by $\lambda_i$, the dynamics of $W$ under gradient flow for a mean squared loss are given by the decomposition

$$W(t) = \sum_{i=1}^{d} v_i w_i(t), \; w_i(t) = e^{-t\lambda_i} w_i(0) + (1 - e^{-\lambda_i t}) w_i^\star. \tag{14}$$

Eq. 14 shows that larger principal components of the data are learned faster than smaller ones. Whitening destroys this hierarchy by setting $\lambda_i = 1 \, \forall i$. If, for example, the data has a simplicity bias (large principal components correspond to signal and small ones correspond to noise), whitening forces the learning algorithm to fit signal and noise directions simultaneously, which results in poorer generalization at finite times during training than would be observed without whitening.

### 2.5 Newton's method is equivalent to training on whitened data for linear least squares models and for wide neural networks

Though in practice unregularized Newton's method is rarely used as an optimization algorithm due to computational complexity, a poorly conditioned Hessian, or poor generalization performance, it serves as the basis of and as a limiting case for most second order methods. Furthermore, in the case of linear least squares models or wide neural networks, it is equivalent to Gauss-Newton descent. In this context, by relating Newton's method to whitening in linear models and wide networks, we are able to give an explanation for why unregularized second order methods have poor generalization performance. We find that our conclusions also hold empirically in a deep CNN (see Figs. 3, 4).

We now compare a pure Newton update step on unwhitened data with a gradient descent update step on whitened data in a linear least squares model. The Newton update step uses the model's Hessian $H$ as a preconditioner for the gradient:

$$W_{\text{Newton}}^{t+1} = W_{\text{Newton}}^t - \eta H^{-1} \frac{\partial L^t}{\partial W^t}. \tag{15}$$

Here we allow for a general step size, $\eta$, with $\eta = 1$ giving the canonical Newton update. It should be noted that this canonical update achieves the optimal solution to the least squares problem in a single step. When $H$ is rank deficient, we take $H^{-1}$ to be a pseudoinverse. For a linear model with mean squared error (MSE) loss, the Hessian is equal to the second moment matrix $F_{\text{train}}$ and the model output evolves as

$$f_{\text{Newton}}^{t+1}(X) = f_{\text{Newton}}^t(X) - \eta \frac{\partial L^t}{\partial f_{\text{Newton}}^t} X_{\text{train}}^\top F_{\text{train}}^{-1} X. \tag{16}$$

We can compare this with the evolution of a linear model $\hat{f}(X) = \hat{W}MX$ trained via gradient descent on whitened data $\hat{X} = MX$ with a mean squared loss:

$$\hat{f}^{t+1}(X) = \hat{f}^t(X) - \eta \frac{\partial L^t}{\partial \hat{f}^t} X_{\text{train}}^\top M^\top M X = \hat{f}^t(X) - \eta \frac{\partial L^t}{\partial \hat{f}^t} X_{\text{train}}^\top F_{\text{train}}^{-1} X. \tag{17}$$

Eqs. 16 and 17 give identical update rules. Thus if both functions are initialized to have the same output, Newton updates give the same predictions as gradient descent on whitened data. While this correspondence is known in the literature, we can now use it to say something further, namely that by applying the argument in Section 2.1, we expect Newton's method to produce linear models that generalize poorly. Our theory results for second order optimization assume a mean squared loss, but we find experimentally that generalization is also harmed with a cross entropy loss in Fig. 3(d).

Finally, many neural network architectures, including fully connected and convolutional architectures, behave as linear models in their parameters throughout training in the large width limit (Lee et al., 2019). The large width limit occurs when the number of units or channels in intermediate layers of the network grows towards infinity. Because of this, *second order optimization harms wide neural networks in the same way it harms linear models* (see Appendix C).

## 3 Experiments

Model and task descriptions and detailed methods are given in Appendix E.

**Whitening and second order optimization impair generalization.** In agreement with theory, in Figs. 3(a) and (b), linear models and MLPs trained on fully whitened data generalize at chance levels until the size of the dataset exceeds the dimensionality of the data, and models trained on train-whitened data perform strictly worse than those trained on unwhitened data. Furthermore, the generalization ability of these models recovers only *gradually* as the dataset grows. On CIFAR-10, a 20% gap in performance between MLPs trained on whitened and unwhitened data persists even at the largest dataset size, suggesting that whitening can remain detrimental even when the number of training examples exceeds the number of features by an order of magnitude.

In Fig. 3(c) we see a generalization gap in the high dimensional regime between WRNs trained on

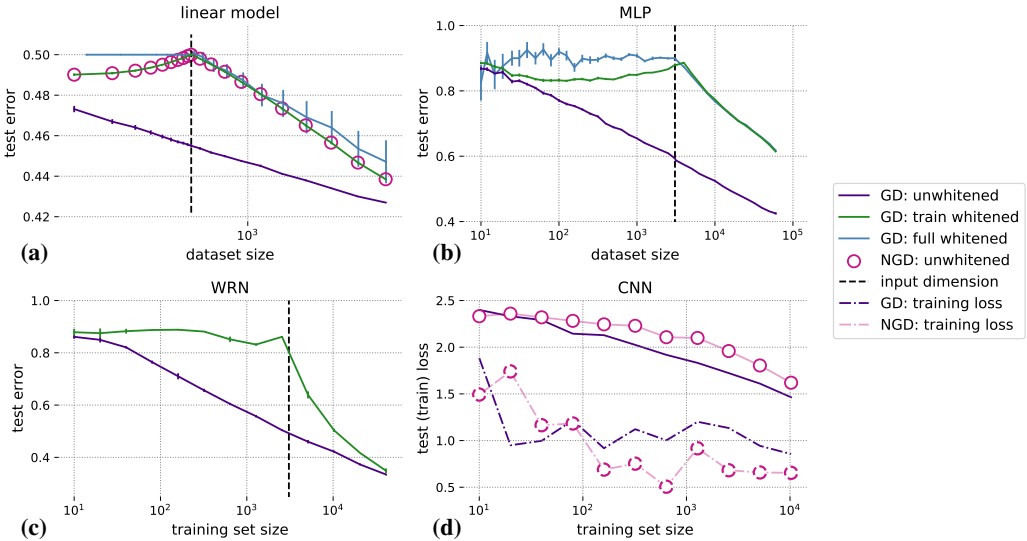

Figure 3: **Whitening and second order optimization reduce or prevent generalization.** *(a)-(c)* Models trained on both fully whitened data (blue; panes a,b) and train-whitened data (green; panes a-c) consistently underperform models trained by gradient descent on unwhitened data (purple; all panes). In (a), Newton's method on unwhitened data (pink circles) behaves identically to gradient descent on whitened data. *(d)* Second order optimization in a convolutional network results in poorer generalization properties than steepest descent. Points plotted correspond to the learning rate and training step with the best validation loss for each method. Data for this experiment was unwhitened. CIFAR-10 is used for all experiments (see Appendix D for experiments on MNIST). In (c) and (d) we use a cross entropy loss (see Appendix E for details).

train-whitened versus unwhitened data, which persists when the size of the dataset grows beyond its dimensionality. This is despite the fact that the convolutional input layer violates the requirements of our theory, and that we used a Xavier initialization scheme, therefore also violating the theoretical requirement for an isotropic weight initialization. We note that these results are consistent with the whitening experiments in the original WRN paper (Zagoruyko & Komodakis, 2016). Generalization ability begins to recover before the size of the training set reaches its input dimensionality, suggesting that the effect of whitening can be countered by engineering knowledge of the data statistics into the model architecture.

In Fig. 3(a), we demonstrate experimentally the correspondence we proved in Section 2.5. In Fig. 3(d), we observe that pure second order optimization similarly harms generalization even in a convolutional network. Despite training to lower values of the training loss, a CNN trained with an unregularized Gauss-Newton method exhibits higher test loss (at the training step with best validation loss) than the same model trained with gradient descent.

**Whitening and second order optimization accelerate training**  In Figs. 4(a) and App.1, linear models trained on whitened data or with a second order optimizer converge to their final loss faster than models trained on unwhitened data, but their best test performance is always worse. In Fig. 4(b), MLPs trained on whitened CIFAR-10 data take fewer epochs to reach to the same training accuracy cutoff than models trained on unwhitened data, except at very small ($< 50$) dataset sizes. The effect is stark at large dataset sizes, where the gap in the number of training epochs is two orders of magnitude large. Second order optimization similarly speeds up training in a convolutional network. In Fig. 4(c), unregularized Gauss-Newton descent achieves its best validation loss two orders of magnitude faster (as measured in the number of training steps) than gradient descent.

**Regularized second order optimization can simultaneously accelerate training and improve generalization.**  In Fig. 5 we perform full batch second order optimization with preconditioner $((1-\lambda)B + \lambda I)^{-1}$, where $\lambda \in [0,1]$ is a regularization coefficient, and $B^{-1}$ is the unregularized Gauss-Newton preconditioner. $\lambda = 0$ corresponds to unregularized Gauss-Newton descent, while $\lambda = 1$ corresponds to full batch steepest descent. At all values of $\lambda$, regularized Gauss-Newton

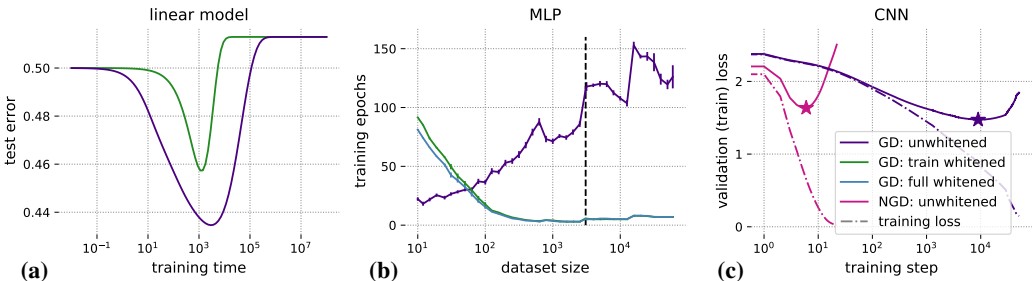

Figure 4: **Models trained on whitened data or with second order optimizers converge faster.** *(a)* Linear models trained on whitened data optimize faster, but their best test accuracy was always worse. Data plotted here is for a training set of size 2560. Similar results for smaller training set sizes are given in Fig. App.1. *(b)* Whitening the data significantly lowers the number of epochs needed to train an MLP to a fixed cutoff in training accuracy, when the learning rate and all other training parameters are kept constant. Discrete jumps in the plot data correspond to points at which the (constant) learning rate was changed. See Appendix E for details. *(c)* Second order optimization accelerates training on unwhitened data in a convolutional network, compared to gradient descent. Data shown is for a training set of size 10240. Stars correspond to values of the validation loss at which test and training losses are plotted in Fig. 3(d).

achieves its lowest validation loss in fewer training steps than steepest descent (Fig. 5(b)). For some values of $\lambda$, the regularized Gauss-Newton method additionally produces lower test loss values than steepest descent (Fig. 5(a)).

Writing the preconditioner in terms of the eigenvectors, $\hat{e}_i$, and eigenvalues, $\lambda_i$ of $B$

$$((1 - \lambda)B + \lambda I)^{-1} = \sum_i \frac{1}{(1 - \lambda)\lambda_i + \lambda} \hat{e}_i \hat{e}_i^T , \qquad (18)$$

we see that regularized Gauss-Newton optimization acts similarly to unregularized Gauss-Newton in the subspace spanned by eigenvectors with eigenvalues larger than $\lambda/(1 - \lambda)$, and similarly to steepest descent in the subspace spanned by eigenvectors with eigenvalues smaller than $\lambda/(1 - \lambda)$. We therefore suggest that regularized Gauss-Newton should be viewed as discarding information in the large-eigenvector subspace, though our theory does not formally address this case. As $\lambda$ increases from zero to one, the ratio $\lambda/(1 - \lambda)$ increases from zero to infinity. Regularized Gauss-Newton method therefore has access to information about the relative magnitudes of more and more of the principal components in the data as $\lambda$ grows larger. We interpret the improved test performance with regularized Gauss-Newton at about $\lambda = 0.5$ in Fig. 5(a) as suggesting that this loss of information within the leading subspace is actually beneficial for the model on this dataset, likely due to aspects of the model's inductive bias which are actively harmful on this task.

## 4 DISCUSSION

**Are whitening and second order optimization a good idea?** Our work suggests that whitening and second order optimization come with costs – a likely reduction in the best achievable generalization. However, both can drastically decrease training time – an effect we also see in our experiments. As compute is often a limiting factor on performance (Shallue et al., 2018), there are many scenarios where faster training may be worth the reduction in generalization. Additionally, the negative effects may be largely resolved if the whitening transform or second order preconditioner are regularized, as is often done in practice (Grosse & Martens, 2016). We observe benefits from regularized second order optimization in Fig. 5, and similar results have been observed for whitening (Lee et al., 2020).

**Directions for future work.** The practice of whitening has, in the machine learning community, largely been replaced by batch normalization, for which it served as inspiration (Ioffe & Szegedy, 2015). Studying connections between whitening and batch normalization, and especially understanding the degree to which batch normalization destroys information about the data distribution, may be particularly fruitful. Indeed, some results already exist in this direction (Huang et al., 2018).

Most second order optimization algorithms involve regularization, structured approximations to the

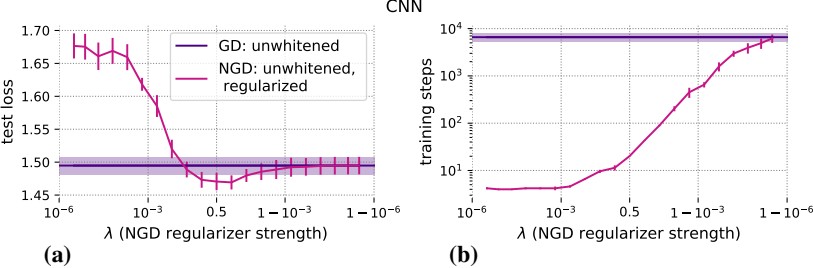

Figure 5: **Regularized second order methods can train faster than gradient descent, with minimal or even positive impact on generalization.** Models were trained on a size 10240 subset of CIFAR-10 by minimizing a cross entropy loss. Error bars indicate twice the standard error in the mean. *(a)* Test loss as a function of regularizer strength. At intermediate values of $\lambda$, the second order optimizer produces *lower* values of the test loss than gradient descent. Test loss is measured at the training step corresponding to the best validation performance for both algorithms. See text for further discussion. *(b)* At all values of $\lambda < 1$, the second order optimizer requires fewer training steps to achieve its best validation performance.

Hessian, and often non-stationary online approximations to curvature. Understanding the implications of our theory results for practical second order optimization algorithms should prove to be an extremely fruitful direction for future work. It is our suspicion that more mild loss of information about the training inputs will occur for many of these algorithms.

Recent work analyzes deep neural networks through the lens of information theory (Tishby & Zaslavsky, 2015; Bassily et al., 2017; Banerjee, 2006; Shwartz-Ziv & Tishby, 2017; Achille & Soatto, 2017; Achille & Soatto, 2019; Amjad & Geiger, 2018; Saxe et al., 2019; Kolchinsky et al., 2018; Alemi et al., 2016; Schwartz-Ziv & Alemi, 2019), often computing measures of mutual information similar to those we discuss. Our result that the only usable information in a dataset is contained in its sample-sample second moment matrix $K$ may inform or constrain this type of analysis.

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

## A    ISOTROPY OF WEIGHT INITIALIZATION IMPLIES CONDITIONAL INDEPENDENCE

In this section we show that for isotropic initial weight distributions,

$$P(W^0 R) = P(W^0) \, \forall \, R \in O(d) \,, \tag{19}$$

the training activations $Z_{\text{train}}$ depend on the training data $X_{\text{train}}$ only through the second moment matrix $K_{\text{train}}$. This is summarized in Eq. 7 repeated here for convenience:

$$\mathcal{I}(Z^0_{\text{train}}; X_{\text{train}} \mid K_{\text{train}}) = 0.$$

The argument is as follows, the isotropy of the weight distribution implies that the distribution of first layer activations conditioned on the training data is invariant under orthogonal transformations.

$$P(Z^0_{\text{train}}|RX_{\text{train}}) = P(Z^0_{\text{train}}|X_{\text{train}}) \, \forall R \in O(d) \,. \tag{20}$$

To derive this we can write the distribution over $Z^0_{\text{train}}$ in terms of the distribution over initial weights, $P(Z^0_{\text{train}}|X_{\text{train}}) = \int DW^0 P(W^0)\delta(Z^0_{\text{train}} - W^0 X_{\text{train}})$. Here $DW^0$ is the uniform measure over the components of $W^0$, $DW^0$. We then have

$$
\begin{aligned}
P(Z^0_{\text{train}}|RX_{\text{train}}) &= \int DW^0 P(W^0)\delta(Z^0_{\text{train}} - W^0 R X_{\text{train}}) \\
&= \int D\tilde{W}^0 P(\tilde{W}^0 R^T)\delta(Z^0_{\text{train}} - \tilde{W}^0 X_{\text{train}}) \\
&= \int D\tilde{W}^0 P(\tilde{W}^0)\delta(Z^0_{\text{train}} - \tilde{W}^0 X_{\text{train}}) \; = \; P(Z^0_{\text{train}}|X_{\text{train}}) \,.
\end{aligned}
\tag{21}
$$

Here, $\delta$ denotes the Dirac delta function. To arrive at the second line we defined $\tilde{W}^0 := W^0 R$ and used the invariance of the measure $DW^0$. The third line follows from the $O(d)$ invariance of the initial weight distribution. Now that we have established the rotational invariance of the distribution over first layer activations we can derive Eq. 7.

By the first fundamental theorem of invariant theory (Kraft & Procesi, 1996), the only $O(d)$ invariant functions of $n$ vectors in $d$ dimensions are the $n^2$ inner products $K_{\text{train}} = X_{\text{train}}^\top X_{\text{train}}$. Thus $P(Z^0_{\text{train}}|X_{\text{train}}) = h(K_{\text{train}})$ for some function $h$, and $P(Z^0_{\text{train}}|X_{\text{train}}, K_{\text{train}}) = P(Z^0_{\text{train}}|K_{\text{train}})$. Eq. 7 then follows from the definition of conditional mutual information.

$$
\begin{aligned}
I(Z^0_{\text{train}}; X_{\text{train}} \mid K_{\text{train}}) :&= \mathbb{E}_{K_{\text{train}}} \left[ D_{\text{KL}}(P(Z^0_{\text{train}}, X_{\text{train}}|K_{\text{train}})||P(Z^0_{\text{train}}|K_{\text{train}})P(X_{\text{train}}|K_{\text{train}})) \right] \\
&= \mathbb{E}_{K_{\text{train}}} \left[ P(X_{\text{train}}|K_{\text{train}})D_{\text{KL}}(P(Z^0_{\text{train}}|X_{\text{train}}, K_{\text{train}})||P(Z^0_{\text{train}}|K_{\text{train}})) \right] \\
&= 0 \,.
\end{aligned}
\tag{22}
$$

## B    WHITENING IN LINEAR MODELS

Linear models are widely used for regression and prediction tasks and provide an instructive laboratory to understand the effects of data whitening. Furthermore, linear models provide additional intuition for why whitening is harmful – whitening puts signal and noise directions in the data second moment matrix, $F$, on equal footing (see Fig. 1). For data with a good signal to noise ratio, unwhitened models learn high signal directions early during training and only overfit to noise at late times. For models trained on whitened data, the signal and noise directions are fit simultaneously and thus the models overfit immediately.

Consider a linear model with mean squared error loss,

$$f(X) = WX \,, \quad L = \frac{1}{2}||f(X) - Y||^2 \,. \tag{23}$$

This loss function is convex. Here we focus on the low dimensional case, $d < n$, where the loss has a unique global optimum $W^\star = F^{-1}_{\text{train}} X_{\text{train}} Y_{\text{train}}$. The model predictions at this global optimum, $f_\star(X) = W^\star X$, are invariant under any whitening transform (2.0.1). As a result, any quality metric

(loss, accuracy, etc...) for this global minimum is the same for whitened and unwhitened data.

The story is more interesting, however, during training. Consider a model trained via gradient flow (similar statements can be made for gradient descent or stochastic gradient descent). The dynamics of the weights are given by

$$\frac{dW}{dt} = -\frac{\partial L}{\partial W}, \quad W(t) = e^{-tF_{\text{train}}} W(0) + (1 - e^{-tF_{\text{train}}}) W^{\star}. \tag{24}$$

The evolution in Eq. 24 implies that the information contained in the trained weights $W(t)$ about the training data $X$ is entirely determined by $F$ and $W^{\star}$. In terms of mutual information, we have

$$\mathcal{I}(W(t); X | F_{\text{train}}, W^{\star}) = 0. \tag{25}$$

As whitening sets $\hat{F}_{\text{train}} = I$, a linear model trained on whitened data does not benefit from the information in $F_{\text{train}}$.

At a more microscopic level, we can decompose Eq. 24 in terms of the eigenvectors, $v_i$, of $F$:

$$W(t) = \sum_{i=1}^{d} v_i w_i(t), \ w_i(t) = e^{-t\lambda_i} w_i(0) + (1 - e^{-\lambda_i t}) w_i^{\star}. \tag{26}$$

We see that for unwhitened data the eigen-modes with larger eigenvalue converge more quickly towards the global optimum, while the small eigen-directions converge slowly. For centered $X$, $F$ is the feature covariance and these eigen-directions are exactly the principle components of the data. As a result, training on unwhitened data is biased towards learning the top principal directions at early times. This bias is often beneficial for generalization. Similar simplicity biases have been found empirically in deep linear networks (Saxe et al., 2014) and in deep networks trained via SGD (Rahaman et al., 2018; Ronen et al., 2019) where networks learn low frequency modes before high. In contrast, for whitened data, $\hat{F}_{\text{train}} = I$ and the evolution of the weights takes the form

$$\hat{w}_i(t) = e^{-t} \hat{w}_i(0) + (1 - e^{-t}) \hat{w}_i^{\star}. \tag{27}$$

All hierarchy between the principle directions has been removed, thus training fits all directions at a similar rate. For this reason linear models trained on unwhitened data can generalize significantly better at finite times than the same models trained on whitened data. Empirical support for this in a linear image classification task with random features is shown in Fig. 3(a).

## B.1 THE GLOBAL OPTIMUM

At the global optimum, $W^{\star} = F_{\text{train}}^{-1} X_{\text{train}} Y_{\text{train}}$, the network predictions on test points can be written in a few equivalent ways,

$$f_{\star}(X_{\text{test}}) = Y_{\text{train}}^T X_{\text{train}}^T F_{\text{train}}^{-1} X_{\text{test}} = Y_{\text{train}}^T K_{\text{train}}^+ K_{\text{train} \times \text{test}} = Y_{\text{train}}^T \hat{K}_{\text{train} \times \text{test}}. \tag{28}$$

Here, the + superscript is the pseudo-inverse, and $\hat{K}_{\text{train} \times \text{test}}$ is the whitened train-test data-data second moment matrix. These expressions make manifest that the test predictions at the global optimum only depend on the training data through $K_{\text{train}}$ and $K_{\text{train} \times \text{test}}$.

## B.2 HIGH DIMENSIONAL LINEAR MODELS

The discussion is very similar in the high dimensional case, $d > n$. In this case, there is no longer a unique solution to the optimization problem, but there is a unique optimum within the span of the data.

$$W_{\parallel}^{\star} = \left( F_{\text{train}}^{\parallel} \right)^{-1} X_{\text{train}}^{\parallel} Y_{\text{train}}, \quad W_{\perp}^{\star} = W_{\perp}(0). \tag{29}$$

Here, we have introduced the notation $\parallel$ for directions in the span of the training data and $\perp$ for orthogonal directions. Explicitly, if we denote by $V^{\parallel} = \{v_1, v_2, \ldots, v_n\} \in \mathbb{R}^{n \times d}$ the non-null eigenvectors of $F_{\text{train}}$ and $V^{\perp} = \{v_{n+1}, v_{n+2}, \ldots, v_d\} \in \mathbb{R}^{(d-n) \times d}$ the null eigenvectors, then $X_{\text{train}}^{\parallel} := V^{\parallel} X_{\text{train}}$, $W_{\parallel} := W V^{\parallel}$, $W_{\perp} := W V^{\perp}$, and $F_{\text{train}}^{\parallel} := V^{\parallel} F_{\text{train}} (V^{\parallel})^T$.

The model predictions at this optimum can be written as

$$f_\star(X_{\text{test}}) = f^0(X_{\text{test}}) - \left(f^0(X_{\text{train}}) - Y_{\text{train}}\right)^T K_{\text{train}}^{-1} K_{\text{train} \times \text{test}} . \tag{30}$$

This is the solution found by GD, SGD, and projected Newton's method.

The evolution approaching this optimum can be written (again assuming gradient flow for simplicity) as

$$W_\|(t) = e^{-tF_{\text{train}}^\|} W_\|(0) + (1 - e^{-tF_{\text{train}}^\|})W_\|^*, \quad W_\perp(t) = W_\perp(0). \tag{31}$$

In terms of the individual components, $[W_\|(t)]_i = e^{-t\lambda_i}[W_\|(0)]_i + (1 - e^{-t\lambda_i})[W_\|^*]_i$.

As above, the hierarchy in the spectrum allows for the possibility of beneficial early stopping, while whitening the data results in immediate overfitting.

### B.3 SUPPLEMENTARY EXPERIMENTS WITH LINEAR LEAST SQUARES

In Fig. App.1 we present the same experiment as in Fig. 4(a) at three additional dataset sizes. In all cases, the best test performance achievable by early stopping on whitened data was worse than on unwhitened data.

In Fig. App.2, we study the effect on generalization of using the entire dataset of 60000 CIFAR-10 images to compute the whitening transform regardless of training set size. We call this type of whitening 'distribution whitening' to indicate that we are interested in what happens when the whitening matrix is approaches its ensemble limit.

In Fig. App.3, we compare generalization performance of models trained on whitened versus unwhitened data from two different parameter initializations. Initial distributions with larger variance produce models that generalize worse, but for a fixed initial distribution, models trained on whitened data generally underperform models trained on unwhitened data.

## C   SECOND ORDER OPTIMIZATION OF WIDE NEURAL NETWORKS

Here we consider second order optimization for wide neural networks. In recent years much progress has been made in understanding the dynamics of wide neural networks (Jacot et al., 2018), in particular it has been realized that wide networks trained via GD, SGD or gradient flow evolve as a linear model with static, nonlinear features given by the derivative of the network map at initialization (Lee et al., 2019).

In this section we extend the connection between linear models and wide networks to second order methods. In particular we argue that *wide networks trained via a regularized Newton's method evolve as linear models trained with the same second order optimizer.*

We consider a regularized Newton update step,

$$\theta^{t+1} = \theta^t - \eta \left(\epsilon \mathbf{1} + H\right)^{-1} \frac{\partial L^t}{\partial \theta} . \tag{32}$$

This diagonal regularization is a common generalization of Newton's method. One motivation for such an update rule in the case of very wide neural networks is that the Hessian is necessarily rank deficient, and so some form of regularization is needed.

For a linear model, $f_{\text{linear}}(x) = \theta^\top \cdot g(x)$, with fixed non-linear features, $g(x)$, the regularized newton update rule in weight space leads to the function space update.

$$f_{\text{linear}}^{t+1}(x) = f_{\text{linear}}^t(x) - \eta \sum_{x_a, x_b \in X_{\text{train}}} \Theta_{\text{linear}}(x, x_a) \left(\epsilon \mathbf{1} + \Theta_{\text{linear}}\right)_{ab}^{-1} \frac{\partial L_b}{\partial f_{\text{linear}}} . \tag{33}$$

Here, $\Theta_{\text{linear}}$, is a constant kernel, $\Theta_{\text{linear}}(x, x') = \frac{\partial f}{\partial \theta}^\top \cdot \frac{\partial f}{\partial \theta} = g^\top(x) \cdot g(x')$.

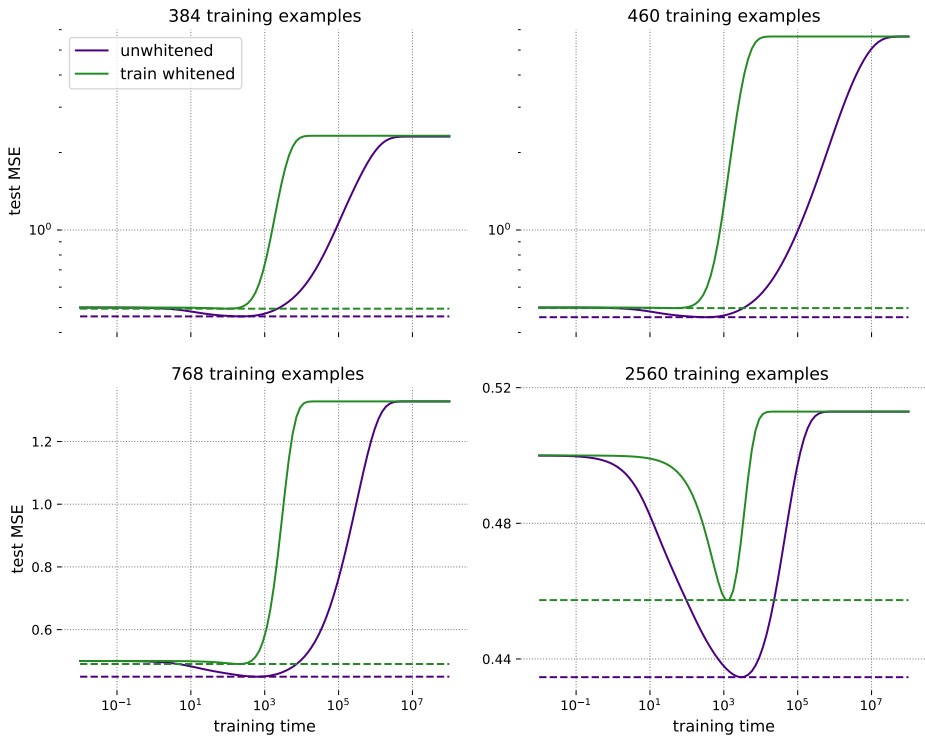

Figure App.1: **Whitening data speeds up training but reduces generalization in linear models.** Here we show representative examples of the evolution of test error with training time in a linear least-squares model where the training set consists of $384, 460, 768, 2560$ examples, as labeled. In all cases, while models trained on train-whitened data (in green) reach their optimal mean squared errors in a smaller number of epochs, they do no better than models trained on unwhitened data (in purple). In the large time limit of training, the two kinds of models are indistinguishable as measured by test error. The $y$-axis in the top row of plots is in log scale for clarity. In all cases, the input dimensionality of the data was $512$.

For a deep neural network, the function space update takes the form.

$$
\begin{aligned}
f^{t+1}(x) = &f^t(x) - \eta \sum_{x_a, x_b \in X_{\text{train}}} \Theta(x, x_a) \left(\epsilon \mathbf{1} + \Theta\right)^{-1}_{ab} \frac{\partial L_b}{\partial f} \\
&+ \frac{\eta^2}{2} \sum_{\mu, \nu = 1}^{P} \frac{\partial^2 f}{\partial \theta_\mu \partial \theta_\nu} \Delta \theta_\mu^t \Delta \theta_\nu^t + \cdots
\end{aligned}
\tag{34}
$$

Here we have indexed the model weights by $\mu = 1 \ldots P$, denoted the change in weights by $\Delta \theta^t$ and introduced the neural tangent kernel (NTK), $\Theta(x, x') = \frac{\partial f^\top}{\partial \theta} \cdot \frac{\partial f}{\partial \theta}$.

In general Eqs. 33 and 34 lead to different network evolution due to the non-constancy of the NTK and the higher order terms in the learning rate. For wide neural networks, however, it was realized that the NTK is constant (Jacot et al., 2018) and the higher order terms in $\eta$ appearing on the second line in vanish at large width (Dyer & Gur-Ari, 2020; Huang & Yau, 2019; Littwin et al., 2020; Andreassen & Dyer; Aitken & Gur-Ari).[2]

With these simplifications, the large width limit of Eq. 34 describes the same evolution as a linear model trained with fixed features $g(x) = \frac{\partial f(x)}{\partial \theta}|_{\theta = \theta_0}$ trained via a regularized Newton update.

---

[2]These simplifications were originally derived for gradient flow, gradient descent and stochastic gradient descent, but hold equally well for the regularized Newton updates considered here. This can be seen, for example, by applying Theorem 1 of (Dyer & Gur-Ari, 2020).

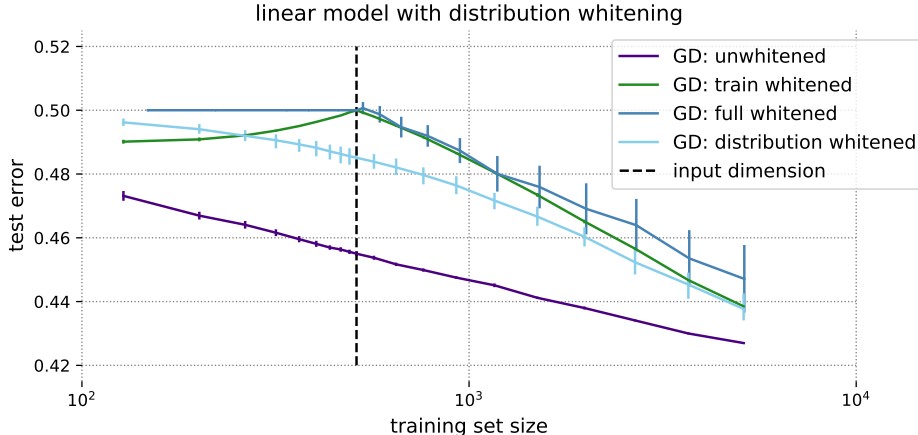

Figure App.2: **Whitening using the entire dataset behaves similarly to conventional whitening, with only a slight improvement in performance.** Whitening using a whitening transform computed on the entire CIFAR-10 dataset of 50000 training and 10000 test images (distribution whitening) improves performance over train and full whitening, but does not close the performance gap with unwhitened data. With the exception of the 'distribution whitened' line, gradient descent data in this plot is identical to Fig. 3(a).

## D   MLP ON MNIST

Here we present the equivalent of Fig. 3(b) for an MLP trained on MNIST. Experimental details are given in Appendix E. Similar to the results in Fig. 3(b) on CIFAR-10, in Fig. App.4, we find that models trained on fully whitened data generalize at chance levels (indicated by a test error of 0.9) in the high dimensional regime. Because MNIST is highly rank deficient, this result holds until the size of the dataset exceeds its input rank. Models trained on train-whitened data also exhibit reduced generalization when compared with models trained on unwhitened data, which exhibit steady improvement in generalization with increasing dataset size.

## E   METHODS

### E.1   MODEL AND TASK DESCRIPTIONS

We describe our basic experiment structure, and follow this with descriptions of the four types of models we studied and associated experimental variations. Details are provided in the rest of this appendix.

The kernel of all our experiments is as follows: From a dataset, we draw a number of subsets, tiling a range of dataset sizes. Each subset is divided into train, test, and validation examples, and three copies are made, two of which are whitened. In one case the whitening transform is computed using only the training examples, and in the other using all the data. Note that the test set size must be reduced in order to run experiments on small datasets, since the test set is considered part of the dataset for full-whitening. Models are trained from random initialization on each of the three copies of the data using the same training algorithm and stopping criterion. Test errors and the number of training epochs are recorded and plotted as functions of dataset size.

**Linear Model.**   We study CIFAR-10 classification learned by optimizing mean squared error loss, where the model outputs are a linear map between the 512-dimensional outputs of a four layer convolutional network at random initialization on CIFAR-10, and their 10-dimensional one-hot labels. This setup is in part motivated by analogy to training the last layer of a deep neural network. We solve the gradient flow equation for the time at which the MSE on the validation set is lowest, and report the test error at that time. The experiment was repeated using Newton's method.

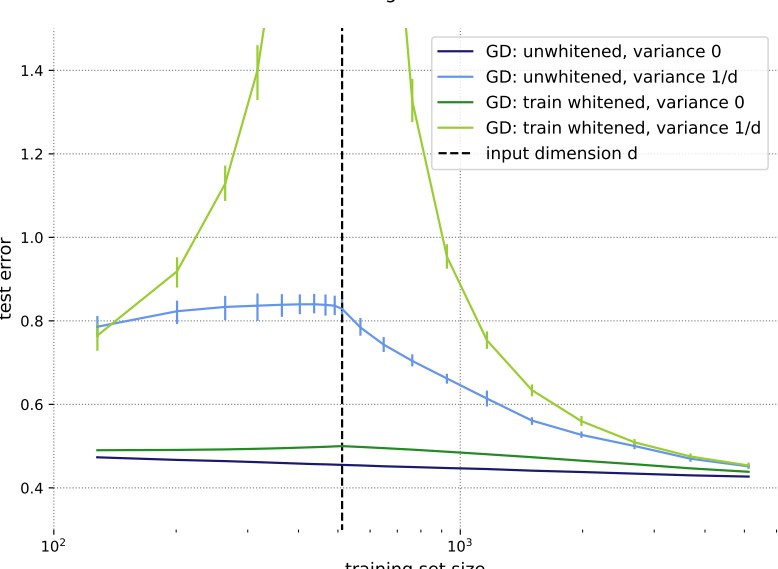

Figure App.3: **The effect of whitening on linear models with non-zero parameter initialization.** Linear models are initialized with parameter variances of 0 or $1/d$. In all cases the test loss is reported for the time during gradient flow training when the model achieves the lowest validation loss. Unwhitened data was scaled to have the same norm accumulated over all samples in the training set as whitened data, for each training set size, to avoid artifacts due to overall input scale. A model output of zero corresponds to a test loss of 0.5. All configurations with loss greater than 0.5 are doing *worse* than an uninformative prediction of 0. At both initialization scales, the model trained on whitened data performs worse than the model trained on unwhitened data for almost all dataset sizes, while for one dataset size they perform similarly. Data for the variance 0 initialization is identical to Fig. 3(a).

**Multilayer perceptron.**   We study fully connected three-layer MLPs on MNIST and CIFAR-10 classification tasks. Training is accomplished using SGD with constant step size until the training accuracy reaches a fixed cutoff threshold, at which point test accuracy is measured.

**Convolutional networks.**   Since our theoretical results on the effect of whitening apply only to models with a fully connected first layer, trained from an isotropic initial weight distribution, we test whether the same qualitative behavior is observed in CNNs trained from a Xavier initialization. We chose the popular wide residual (WRN) architecture (Zagoruyko & Komodakis, 2016), trained on CIFAR-10. Training was performed using full batch gradient descent with a cosine learning rate schedule for a fixed number of epochs. Full batch training was used to remove experimental confounds from choosing minibatch sizes at different dataset sizes. A validation set was split from the CIFAR-10 training set. Test error corresponding to the parameter values with the lowest validation error was reported.

We also trained a smaller CNN (a ResNet-50 convolutional block followed by an average pooling layer and a dense linear layer) on unwhitened data with full batch gradient descent and with the Gauss-Newton method (with and without a scaled identity regularizer) to compare their respective generalization performances. A grid search was performed over learning rate, and step sizes were chosen using a backoff line search initialized at that learning rate. Test and training losses corresponding to the best achieved validation loss were reported. Note that this experiment is relatively large scale; because we perform full second order optimization to avoid confounds due to choosing a quasi-Newton approximation, iterations are cubic in the number of model parameters.

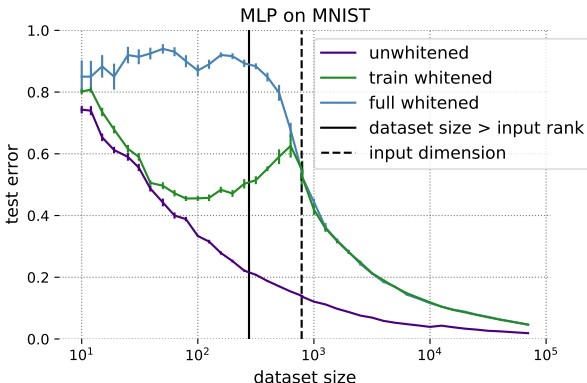

Figure App.4: **Whitening MNIST before training negatively impacts generalization in MLPs.** Models trained on fully whitened data (in blue) are unable to generalize until the size of the dataset exceeds its maximal input rank of 276, indicated by the solid black vertical line. Regardless of how the whitening transform is computed, models trained on whitened data (blue and green) consistently underperform those trained on unwhitened data (in purple).

### E.2  WHITENING METHODS

#### E.2.1  PCA WHITENING

Consider a dataset $X \in \mathbb{R}^{d \times n}$. PCA whitening can be viewed as a two-step operation involving rotation of $X$ into the PCA basis, followed by the normalization of all PCA components to unity. We implement this transformation as follows. First, we compute the the singular value decomposition of the unnormalized feature-feature second moment matrix $XX^\top$:

$$XX^\top = U\Sigma V^\top, \tag{35}$$

where the rank of $\Sigma$ is $\min(n, d)$. The PCA whitening transform is then computed as $M = \Sigma^{-1/2} \cdot V^\top$, where the dot signifies element-wise multiplication between the whitening coefficients, $\Sigma^{-1/2}$, and their corresponding singular vectors. When $\Sigma$ is rank deficient ($n < d$), we use one of two methods to stabilize the computation of the inverse square root: the addition of noise, or manual rank control. In the former, a small jitter is added to the diagonal elements of $\Sigma$ before inverting it. This was implemented in the experiments in linear models. In the latter, the last $d - n$ diagonal elements of $\Sigma^{-1/2}$ are explicitly set to unity. This method was implemented in the MLP experiments.

#### E.2.2  ZCA WHITENING

ZCA (short for zero-phase components analysis) (Bell & Sejnowski, 1997) can be thought of as PCA whitening followed by a rotation back into the original basis. The ZCA whitening transform is $M = U\Sigma^{-1/2} \cdot V^\top$. ZCA whitening produces images that look like real images, preserving local structure. For this reason, it is used in the CNN experiments.

### E.3  LINEAR MODEL

**Dataset composition.** The dataset for this experiment was a modified version of CIFAR-10, where the images were first processed by putting them through an off-the-shelf (untrained) four layer convolutional network with $\tanh$ nonlinearities and collecting the outputs of the penultimate layer. This resulted in a dataset of $512$-dimensional images and their associated labels. Both the CIFAR-10 training and test sets were processed in this way. The linear estimator was trained to predict one-hot (ten dimensional) labels.

Training set sizes ranged from $128$ to $5120$ examples, randomly sampled from the preprocessed CIFAR-10 data. For experiments on unwhitened and train-whitened data, a validation set of $10000$ images was split from the CIFAR-10 training set, and test error was measured on the CIFAR-10 test

set. For experiments on fully whitened data, validation and test sets of 10 images each were split from the CIFAR-10 training and test sets, respectively.

**Whitening.** At each training set size, four copies of the data were made, and three were whitened using the PCA whitening method. For train-whitened data, the whitening transform was computed using only the training examples. For fully whitened data, the twenty validation and test images were included in the computation of the whitening transform. For distribution whitened data (Fig. App.2), the entire CIFAR-10 dataset of 60000 images (train as well as test) was used to compute the whitening transform.

**Training and Measurements.** We used a mean squared error loss function. Weights were initialized to all-zeros, except for the data in Fig. App.3, for which initial weights were drawn from a Gaussian with variance $1/d$. At each training set size, fifty models (initialized with different random seeds) were trained with full-batch gradient descent, with the optimization path defined by the gradient flow equation. Writing the model parameters as $\phi$, this equation is

$$\phi(t) = \phi^* + e^{-tCB}(\phi^* - \phi^{(0)})$$

for preconditioner $B$, feature-feature correlation matrix $C$, infinite-time solution $\theta^*$, and initial iterate $\theta^{(0)}$. In the case of gradient descent, the preconditioner $B$ is simply the identity matrix.

In order to generate the plot data for Fig. 3(a), we solved the gradient flow equation for the parameters $\phi$ that achieved the lowest validation error, and calculated the test error achieved by those parameters. Mean test errors and their inner 80th percentiles across the twenty different initializations and across whitening states were computed and plotted. To make the plots in Fig. 4(a) and App.1, we tracked test performance over the course of training on unwhitened and train-whitened data.

On train-whitened datasets, we also implemented Newton's Method. This was done by putting the preconditioner $B$ in the gradient flow equation equal to the inverse Hessian, i.e. $\left(XX^\top\right)^{-1}$. The preconditioner was computed once using the whole training set, and remained constant over the course of training. For experiments on whitened data, the data was whitened before computing the preconditioner.

### E.4 MULTILAYER PERCEPTRON

### E.4.1 ON MNIST

**Architecture.** We used a $784 \times 512 \times 512 \times 10$ fully connected network with a rectified linear nonlinearity in the hidden layers and a softmax function at the output layer. Initial weights were sampled from a normal distribution with variance $10^{-4}$.

**Dataset composition.** The term "dataset size" here refers to the total size of the dataset, i.e. it counts the training as well as test examples. We did not use validation sets in the MLP experiments. Datasets of varying sizes were randomly sampled from the MNIST training and test sets. Dataset sizes were chosen to tile the available range (0 to 70000) evenly in $\log$ space. The smallest dataset size was 10 and the two largest were 50118 and 70000. For all but the largest size, the ratio of training to test examples was $8 : 2$. The largest dataset size corresponded to the full MNIST dataset, with its training set of 60000 images and test set of 10000 images.

The only data preprocessing step (apart from whitening) that we performed was to normalize all pixel values to lie in the range $[0, 1]$.

**Whitening.** At each dataset size, three copies of the dataset were made and two were whitened. Of these, one was train-whitened and the other fully whitened. PCA whitening was performed. The same whitening transform was always applied to both the training and test sets.

**Training and Measurements.** We used sum of squares loss function. Initial weights were drawn from a Gaussian with mean zero and variance $10^{-4}$. Training was performed with SGD using a constant learning rate and batch size, though these were both modulated according to dataset size. Between a minimum of 2 and a maximum of 200, batch size was chosen to be a hundredth of the number of training examples. We chose a learning rate of $0.1$ if the number of training examples was $\leq 50$, $0.001$ if the number of training examples was $\geq 10000$, and $0.01$ otherwise. Models were trained to $0.999$ training accuracy, at which point the test accuracy was measured, along with

the number of training epochs, and the accuracy on the full MNIST test set of $10000$ images. This procedure was repeated twenty times, using twenty different random seeds, for each dataset. Means and standard errors across random seeds were calculated and are plotted in Fig. App.4.

For example, at the smallest dataset size of $10$, the workflow was as follows. Eight training images were drawn from the MNIST training and two as test images were drawn from the MNIST test set. From this dataset, two more datasets were constructed by whitening the images. In one case the whitening transform was computed using only the eight training examples, and in another by using all ten images. Three copies of the MLP were initialized and trained on the eight training examples of each of the three datasets to a training accuracy of $0.999$. Once this training accuracy was achieved, the test accuracy of each model on the two test examples, and on the full MNIST test set, was recorded, along with the number of training epochs. This entire procedure was repeated twenty times.

**Computation of the input rank of MNIST data.** MNIST images are encoded by $784$ pixel values. However, the input rank of MNIST is much smaller than this. To estimate the maximal input rank of MNIST, at each dataset size (call it $n$) we constructed twenty samples of $n$ images from MNIST. For each sample, we computed the $784 \times 784$ feature-feature second moment matrix $F$ and its singular value decomposition, and counted the number of singular values larger than some cutoff. The cutoff was $10^{-5}$ times the largest singular value of $F$ for that sample. We averaged the resulting number, call it $r$, over the twenty samples. If $r = n$, we increased $n$ and repeated the procedure, until we arrived at the smallest value of $n$ where $r > n$, which was $276$. This is what we call the maximal input rank of MNIST, and is indicated by the solid black line in the plot in Appendix D.

### E.4.2   ON CIFAR-10

The procedure for the CIFAR-10 experiments was almost identical to the MNIST experiments described above. The differences are given here.

The classifier was of shape $3072 \times 2000 \times 2000 \times 10$ - slightly larger in the hidden layers and of necessity in the input layer.

The learning rate schedule was as follows: $0.01$ if the number of training examples was $\leq 504$, then dropped to $0.005$ until the number of training examples exceeded $2008$, then dropped to $0.001$ until the number of training examples exceeded $10071$, and $0.0005$ thereafter.

The CIFAR-10 dataset is full rank in the sense that the input rank of any subset of the data is equal to the dimensionality, $3072$, of the images.

### E.4.3   FIG. 3(B), APP.4 PLOT DETAILS

In Figs. 3(b) and App.4, for models trained on unwhitened data and train-whitened data, we have plotted test error evaluated on the full CIFAR-10 and MNIST test sets of $10000$ images. For models trained on fully whitened data, we have plotted the errors on the test examples that were included in the computation of the whitening transform.

### E.5   CONVOLUTIONAL NETWORKS

### E.5.1   WRN

**Architecture.** We use the ubiquitous Wide ResNet 28-10 architecture from (Zagoruyko & Komodakis, 2016). This architecture starts with a convolutional embedding layer that applies a $3 \times 3$ convolution with 16 channels. This is followed by three "groups", each with four residual blocks. Each residual block features two instances of a batch normalization layer, a convolution, and a ReLU activation. The three block groups feature convolutions of 160 channels, 320 channels, and 640 channels, respectively. Between each group, a convolution with stride 2 is used to downsample the spatial dimensions. Finally, global-average pooling is applied before a fully connected readout layer.

**Dataset composition.** We constructed thirteen datasets from subsets of CIFAR-10. The thirteen training sets ranged in size from 10 to 40960, and consisted of between $2^0$ and $2^{12}$ examples per class. In addition, we constructed a validation set of 5000 images taken from the CIFAR-10 training set which we used for early stopping. Finally, we use the standard CIFAR-10 test set to report

performance.

**Whitening.** We performed ZCA whitening using only the training examples to compute the whitening transform.

**Training and Measurements.** We used a cross entropy loss and the Xavier weight initialization. We performed full-batch gradient descent training with a learning rate of $10^{-3}$ until the training error was less than $10^{-3}$. We use TPUv2 accelerators for these experiments and assign one image class to each chip. Care must be taken to aggregate batch normalization statistics across devices during training. After training, the test accuracy at the training step corresponding to the highest validation accuracy was reported. At each dataset size, this procedure was repeated twice, using two different random seeds. Means and standard errors across seeds were calculated and are plotted in Fig. 3(c).

### E.5.2 CNN

**Architecture.** The network consisted of a single ResNet-50 convolutional block followed by a flattening operation and two fully connected layers of sizes 100 and 200, successively. Each dense layer had a $tanh$ nonlinearity and was followed by a layer norm operation.

**Dataset composition.** Training sets were of eleven sizes: 10, 20, 40, 80, 160, 320, 640, 1280, 2560, 5120, and 10240 examples. A validation set of 10000 images was split from the CIFAR-10 training set.

**Training and Measurements.** We used a cross entropy loss. Initial weights were drawn from a Gaussian with mean zero and variance $10^{-4}$. Training was accomplished once with the Gauss-Newton method (see (Botev et al., 2017) for details), once with full batch gradient descent, and once with a regularized Gauss-Newton method. With a regularizer $\lambda \in [0, 1]$, the usual preconditioning matrix $B$ of the Gauss-Newton update was modified as $((1 - \lambda)B + \lambda I)^{-1}$. This method interpolates between pure Gauss-Newton ($\lambda = 0$) and gradient descent ($\lambda = 1$). In the Gauss-Newton experiments, we used conjugate gradients to solve for update directions; the sum of residuals of the conjugate gradients solution was required to be at most $10^{-5}$.

For the gradient descent and unregularized Gauss-Newton experiments, at each training set size, ten CNNs were trained beginning with seven different initial learning rates: $2^{-8}, 2^{-6}, 2^{-4}, 2^{-2}, 1, 4,$ and 16. After the initial learning rate, backtracking line search was used to choose subsequent step sizes. Models were trained until they achieved 100% training accuracy. The model with the initial learning rate that achieved the best validation performance of the seven was then selected. Its test performance on the CIFAR-10 test set was evaluated at the training step corresponding to its best validation performance. The entire procedure was repeated for five random seeds. In Fig. 3(d), we have plotted average test and validation losses over the random seeds as functions of dataset size and training algorithm. In Fig. 4(c), we have plotted an example of the validation and training performance trajectories over the course of training for a training set of size 10240.

For the regularized Gauss-Newton experiment, the only difference is that we trained one CNN at each initial learning rate per random seed, and then selected the model with the best validation performance. In Fig. 5, we have plotted average metrics over the five random seeds. Errorbars and shaded regions indicate twice the standard error in the mean.

