# OpenReview forum: "Whitening and second order optimization both destroy information about the dataset, and can make generalization impossible"
_ICLR.cc/2021/Conference — Reject_

### Official Review · AnonReviewer3 · 2020-10-26
**The paper is well-executed and has interesting implilcations, but I have concerns that the results might already exist under different terminology**

**Rating:** 7
**Confidence:** 3

**Review:**

## Overview
---
In supervised learning tasks, it is common in practice to apply a *whitening* transformation to remove correlations between input features. This can improve the conditioning of the underlying data manifold, enabling faster convergence. This paper shows  that for a large class of models --- models $f$ consisting of a fully-connected layer followed by an arbitrary parameterized function, $f(X) = g_\theta(WX)$ --- data whitening removes all information that is relevant for generalization. Furthermore, this has implications for the generalization ability of second-order methods such as Newton's method due to a well-known equivalence between Newton's method and steepest-descent applied to whitened data. The effects suggested by the presented theory are verified empirically, and the are additionally observed in convolutional neural networks, suggesting that the phenomenon could apply more broadly to more complicated connectionist models as well.

**Overall, I recommend that the paper be accepted**. The theoretical results are interesting and potentially high-impact, and the writing clarity was excellent throughout. My main reservation about the paper is that it's not clear to me which insights are being proposed as novel, and among those, which actually are novel. In particular,
- **Section 2.5**: The relationship between Newton's method and data whitening is well-known. Does section 2.5 include any novel insights, or is this included for completeness? If it's the latter it seems like it would be better-placed in an appendix; the fact that the main results relate also to second-order methods could then be noted in the discussion
- **Generally** speaking, how does what's already known from e.g. dimensionality reduction and PCA play into this? As mentioned in the paper, whitening is essentially putting the signal and noise in the data on equal footing. From a principal components perspective, it's as if you're forcing the model to consider all components to be equally predictive, which would clearly harm generalization when the data has strong feature correlations. Is there truly no prior result of this kind in that literature? It seems quite fundamental, and my concern is that this phenomenon might be already well-known under different terminology in another literature.

*(The following question is out of scope but I think could potentially increase the impact of the paper a great deal)* Do you suspect these issues hold also for the online second-order methods, such as Online Newton Step and AdaGrad? If so, adding some discussion about this could potentially increase the impact of this work since AdaGrad (and it's heuristic descendants Adam, RMSProp etc.) are by-far the most common algorithms used for optimizing neural networks in practice

## Clarifications
---
- **Page 4**: W is isotropic, and whitening makes X isotropic, so that Z is isotropic, so intuitively the combination of whitening and isotropic initialization results in trying to make predictions from isotropic noise. Is it specifically the *combination* of whitening and isotropic initialization that's the problem? The paper seems to be really centered around the whitening, when it seems to be the specific combination of the two rather than whitening alone. Or is there a nuance I'm missing here?
- **Page 8**: *"We therefore believe that regularized Gauss-Newton should be viewed as discarding information in the
large-eigenvector subspace"*. I'm not sure I understand what is being said here. Wouldn't this suggest that discarding principle components is beneficial? this seems to run contrary to a lot of the dimensionality reduction literature.


## Minor Comments (which did not influence my score but could improve clarity)
---
- **Page 2**: *"Our result is not restricted to neural networks, and applies to any model in which the input is transformed by a dense matrix multiply with isotropic weight initialization"*
 should read *dense matrix with isotropic weight initialization*?
- **Page 2**: It is not really necessary to list 34 (!!) papers on second-order optimization in a single sentence; there are plenty of surveys on the topic that could be linked instead
- **Page 7, Figure 4**: *"Linear models trained on whitened data optimize faster, but their best test accuracy is always worse"*.
This could use some minor rephrasing to say that the best test accuracy *was* always worse, as this conclusion is about this specific problem rather than for all problems generally
- **Page 8**:
 *"Regularized Gauss-Newton optimization acts similarly to unregularized Gauss-Newton in the subspace spanned by eigenvectors with eigenvalues larger than λ/(1 − λ), and similarly to steepest descent in the subspace spanned by eigenvectors with eigenvalues smaller than λ/(1 − λ)"*
This fact is not immediately obvious, so a citation would be helpful here (or potentially a link to an appendix if it's not a well-known result)
- **Page 14, appendix A**: The integral notation here is unclear to me; what kind of integral is this supposed to be? what is delta?

---

> ### Author Response · Authors · 2020-11-20
> **Response to AnonReviewer3, 2/2**
>
> \> ___Re: minor comments___
>
> We thank the reviewer for their suggestions to improve the clarity of our exposition, and have implemented most of these.
> In Appendix A the measure of integration is the uniform measure over the components of W^{0}. This is defined in the text. It can also be written as the product \prod_i dW^{0}_{i}. The delta is a Dirac delta function.
> We have expanded the discussion on page 8.
>
>
> The primary concern you expressed about our paper was (boldly paraphrasing!) that the results seemed basic and important enough that a subset of them must surely already be known and published. We hope you are more convinced of the work’s novelty after our rebuttal, and after seeing that no other reviewer raised concerns about missing related work. If you are now more convinced of the paper’s novelty, we respectfully ask you to consider more strongly supporting acceptance, and raising your score accordingly.
>
> Thank you again!

---

> > ### Comment · AnonReviewer3 · 2020-11-23
> > **Follow-up**
> >
> > Thanks for addressing all of my concerns and the detailed response. I've decided to increase my score.

---

> ### Author Response · Authors · 2020-11-20
> **Response to AnonReviewer3, 1/2**
>
> Thank you for your careful review and specific questions!
>
> \> ___The relationship between Newton's method and data whitening is well-known. Does section 2.5 include any novel insights, or is this included for completeness?___
>
> The novelty in Section 2.5 is in using the known relationship between data whitening and Newton’s method to apply our results on the effect of data whitening on generalization to Newton’s method. We have edited the text to make this point more clear.
>
> (We do believe our observation that Newton’s method on a wide neural network is equivalent to Newton’s method on a linear model is novel, but this statement follows directly from previous work and is not a primary contribution.)
>
> \> ___Generally speaking, how does what's already known from e.g. dimensionality reduction and PCA play into this? ... Is there truly no prior result of this kind in that literature?___
>
> It is difficult to prove a negative, but we were not able to find references looking at the effect of data whitening on generalization in machine learning models, nor have any reviewers or peers who have seen our work brought any prior work on this to our attention.
>
> We did find prior work examining the information loss due to the *dimensionality reduction* in PCA (see Geiger and Kubin, 2012). However, that is a different mechanism for information loss -- our analysis holds for fixed dimensionality.
>
> Practitioners do often regularize whitening transforms and second order optimizers, so the effect may be implicitly known, or familiar but never formalized, in some communities. If this is the case, we believe formalizing and publishing it is an important contribution.
>
> \> ___Do you suspect these issues hold also for the online second-order methods, such as Online Newton Step and AdaGrad?___
>
> This is an excellent question! Our suspicion is that qualitatively similar, but more mild, effects hold for approximate second order methods like those you list. However, we have not yet explored this theoretically or experimentally, and so we cannot state this with confidence. Based on your feedback, we have added a brief discussion of this to our Discussion section.
>
> \> ___Re: clarifications___
>
> Page 4: Indeed, it is the combination of isotropic weight initialization and whitening the data that is problematic. However, since weight initialization is usually from an isotropic distribution, we believe that this requirement is not a significant restriction.
>
>
> Page 8: For ease of discussion, let’s divide the eigenvalues of the feature second moment matrix into two subsets. The subset of eigenvalues larger than lambda/(1 \- lambda) we will call L, and the subset of eigenvalues smaller than this cutoff we will call S. Regularized Gauss-Newton method behaves like unregularized Gauss-Newton method in the subset of feature space spanned by the eigenvectors corresponding to L, and behaves like gradient descent in the subset of features space spanned by eigenvectors corresponding to S. We have already established that the unregularized Gauss-Newton method underperforms gradient descent on generalization due to its connection with whitening, and that this effect is driven by a loss of information. Therefore we can view the regularized Gauss-Newton method as being insensitive to information contained in the subset of feature space spanned by the eigenvectors corresponding to L. Thanks to your feedback we have expanded the discussion in the text regarding this point.
>
> \> ___Wouldn't this suggest that discarding principle components is beneficial? this seems to run contrary to a lot of the dimensionality reduction literature.___
>
> There is a key nuance, which is that regularized second order optimization as described here will discard information about the relative magnitudes of the leading principal components. This is distinct from discarding those components entirely, and so results from the dimensionality literature do not directly apply. When the dataset size is smaller than the number of components larger than the cutoff of lambda/(1-lambda), it will still be possible to generalize from those components via interpolation (see Equation 13). When the dataset size is larger than the number of components larger than the cutoff, it will further be possible to learn from residual sample-sample second order information in those dimensions.
>
> However, we also found these experimental results surprising, as it is counterintuitive that discarding information will lead to improved generalization! We have expanded our discussion of regularized Gauss-Newton in Section 3.

---

### Official Review · AnonReviewer4 · 2020-10-28
**An interesting analysis of why whitening and second order methods usually lead to worse generalization than first order methods.**

**Rating:** 7
**Confidence:** 4

**Review:**

Summary

The authors analyse the training dynamics of a machine learning model consisting in a linear unit, followed by any parametrized function. The authors in particular focus on the impact of whitening  the data beforehand or using second order methods. They show that the learned parameters of the model only depend on the training data through its Gram matrix. Since whitening trivializes the Gram matrix, the authors argue that whitening destroys important information.

Major comments
- The article is well written, the arguments are well presented and easily understood.
- The main message of the paper, contained in Fig. 3, is easily reproduced with a few lines of code.
- The paper sheds an interesting light on the generalization properties of second order methods.
Thm.2.2.1 and 2.2.2 treat the initialization of the weights as random variables. I believe that it makes them rather useless for practical applications, because in practice the machine learning model will only be initialized once. Of course, the theorems also apply if we assume that the initial weights are zero (or close to zero), which is still a very interesting case. In the experiments description of fig.3. it would be worthwhile to better describe the initialization strategy for the linear weights. For instance, I used as X the load_digits dataset in scikit-learn, as Y a one hot encoding of the target, and a simple linear model with MSE loss $f(W) = \|WX - Y\|^2$, and managed to get the same fig.3.a as the authors, but only if I initialize the weights very close to 0, or at 0.
- I think that Sec.2.4 is too shallow. First, it should be clearly stated that eq.14 only works for MSE loss with a purely linear model. Second, I think that another important concept which is not mentioned in the article is that early stopping is similar to regularization of the parameters, which is why the reported ‘test error’ in the experiments is lower than the test error obtained by perfectly minimizing the train error. See e.g. Yao, Yuan, Lorenzo Rosasco, and Andrea Caponnetto. "On early stopping in gradient descent learning." Constructive Approximation 26.2 (2007): 289-315.
- In Sec.2.5, the authors consider Newton’s method with MSE loss on a linear problem, which converges in just one step with $\eta=1$. This should be acknowledged, and maybe it would be easier to defend the point by letting $\eta \to 0$ instead.

Minor comments
- The meaning of ‘test error’ in the figure legend is a bit vague, and I think it should rather be replaced by ‘best test error during training’ or something similar.
- In fig.4.b, does training epochs mean ‘number of epochs to reach the best test error’?

Misc
- What does WRNs mean?

---

> ### Author Response · Authors · 2020-11-20
> **Response to AnonReviewer4**
>
> Thank you for volunteering your time to review our paper, and for your careful review!
>
> \> ___The main message of the paper, contained in Fig. 3, is easily reproduced with a few lines of code.___
>
> It’s very neat that you were able to reproduce our results so easily!
>
> \> ___In the experiments description of Fig.3. it would be worthwhile to better describe the initialization strategy for the linear weights.___
>
> We used a Gaussian with mean zero and variance 1e-4 to initialize weights for all experiments except the linear model, which we initialized with zero weights. We thank the reviewer for catching our inadvertent omission of this information from the Methods (Appendix E), and have now corrected it!
>
> \> ___Thm.2.2.1 and 2.2.2 treat the initialization of the weights as random variables. I believe that it makes them rather useless for practical applications, because in practice the machine learning model will only be initialized once.___
>
> We believe these results are relevant for practical applications. For high dimensional input, the outputs of a model trained on full-whitened data will contain no information about its training inputs. It is possible for a single random initialization of this model to do well, in the same way it would be possible to do well on a test set by randomly flipping a coin to predict each test example. However, *a typical random initialization will not perform any better than would a typical sequence of coin flips!*
>
> As further evidence for the relevance of the results to individual models, we can see in Figure 3 that the performance of randomly initialized models concentrates around their average case performance.
>
> \> ___It should be clearly stated that eq.14 only works for MSE loss with a purely linear model… early stopping is similar to regularization of the parameters.___
>
> Thank you for catching this. We have updated Section 2.4 to clarify that we analyze the mean squared loss. We emphasize though that our results in Sections 2.1 through 2.3 hold for any loss function, not just MSE loss, and that many of our experiments use cross entropy loss.
>
> We have also added a brief discussion of the effect of early stopping, and its relationship to weight regularization, to Section 2.4.
>
> \> ___In Sec.2.5, the authors consider Newton’s method with MSE loss on a linear problem, which converges in just one step with η=1. This should be acknowledged.___
>
> We have updated Section 2.5 with a sentence stating that a single Newton step with $\eta = 1$ solves the optimization problem over the training dataset.
>
> \> ___Re: minor comments and miscellaneous___
>
> Figure 4b. In all experiments with MLPs, the models were trained to a fixed training accuracy cutoff, at which point test error was measured. The label of the y-axis indicates the number of epochs it took to reach this training cutoff. This is explained in the figure caption and in Appendix E.
>
> WRN is the acronym for “wide residual network”.

---

> > ### Comment · AnonReviewer4 · 2020-11-20
> > **Concerns regarding initialization**
> >
> > Thanks for your careful response which addresses most of my points.
> >
> > I am still concerned by the usefulness of the random part of the theorem. In the experiments, the authors initialize the weights with very small variance (much lower than what is usual in practice ?).  What happens when initialized with a variance of 1? What happens when the weights in the linear model are initialized at random with a non-zero variance?

---

> > > ### Author Response · Authors · 2020-11-21
> > > **Re: Concerns regarding initialization**
> > >
> > > We will re-run our linear model experiments with large variance at init, to address your experimental question. We will send another message when those experiments finish.
> > >
> > > In the meantime, I’d like to bring up two points which may be useful for addressing your concerns:
> > >
> > > First, and simplest, point: If a model is initialized with very large variance, performance on both whitened and unwhitened data is likely to be similarly terrible. Is this what you are seeing in your own experiments with linear models?
> > >
> > > Typical initialization variances scale like 1/[# features] (see Xavier, He, Glorot initialization schemes), so initializing a model with variance 1 may cause bad performance.
> > >
> > > The second and more complex point involves how mutual information should be interpreted for single samples: The statement in Equation 11,
> > > I( f_test ; \hat{X} | Y_train ) = 0,
> > > where \hat{X} is whitened data, also implies
> > > I( f_test ; Y_test | Y_train ) = 0,
> > > where the true test labels Y_test are a function of X_test.
> > >
> > > What this second equality means is that if I tell you the prediction of a model on a test point, that provides no information at all about the actual test label. This is true no matter the variance of parameter initialization. The predictions of a model on the test points are indistinguishable from random guesses on those test points.
> > >
> > > It is true that sometimes a random parameter draw will produce a model that does well on Y_test. However this will be exactly as (un)likely as a random parameter draw producing a model that does well on a random permutation of Y_test.
> > >
> > > Additionally, even if a random parameter draw produces a model that does well on the first N test points ... it is exactly as likely to get the N+1 test point wrong as the model from a random parameter draw that got the first N test points wrong*.
> > >
> > > Stay tuned for new experiments on the linear model, and also for an update to the text further clarifying this.
> > >
> > > Thank you again for your work reviewing our paper!
> > >
> > > \* Assuming that both models have the same marginal distribution over test predictions.

---

> > > > ### Author Response · Authors · 2020-11-25
> > > > **Re: concerns regarding initialization**
> > > >
> > > > We have completed a new experiment with the linear model, comparing the performances of models trained on whitened versus unwhitened data across two initial parameter distributions. One is the all-zeros initialization, the other is a Gaussian with variance 1/#features. The data is in Fig. App 3. As we suspected above, we find that a larger initial variance in the weight distribution leads to poor performance regardless of whether or not the data is whitened (both kinds of data produce models that generalize worse than chance). Consistent with predictions made in the paper, for a fixed initial weight distribution, models trained on whitened data generally underperform models trained on unwhitened data.
> > > >
> > > > If you are working to reconcile these results with your own experimental explorations, then one key experimental detail is to make sure the overall scale of the unwhitened data is that same as that of the whitened data. Otherwise, the model trained on the larger-norm dataset will be at an unfair disadvantage.
> > > >
> > > > Note. We made an error in our earlier response about the weight initialization for WRN experiments. We used a Xavier initialization scheme, not a Gaussian with small variance. We have modified the paper to discuss this: we now note in the body that the WRN probes generalization behavior when our theoretical requirements of a fully connected input layer and an isotropic weight initialization are both relaxed. We have additionally corrected this in Appendix E.

---

### Official Review · AnonReviewer2 · 2020-10-28
**Derivation seems suspicious; needs clarification**

**Rating:** 4
**Confidence:** 4

**Review:**

Topic: whitening destroys generalization


main contribution

This work offers a mutual information perspective to explain the relations between whitening/second-order optimization and generalization.

Strength

- The perspective is interesting and a bit intriguing. But many discrepancies may need clarifications.

Weakness

- The title and the sentiment of the paper may be misleading. It is scary at first glance, but the conclusion and experiments do not really support the claim “destroy”. It may be important to keep the title accurate other than eye-catching.  The authors may argue “can” is the key in the title, but the reviewer feels this may be a bit too subtle.

- Conditional independence and generalization. The theorems established a certain conditional independence between training features, the model, and the test data. The conditional independence is derived using gradient descent of the whitened data. This is understandable, but the implication on generalization is not crystal clear.

- The claims may need more explanation and be better represented; some simple examples may suggest different conclusion. For example, the reviewer considered a simple least squares problem

min_theta ||y-X*theta||^2 with whitened X

the solution is theta_opt = X^T*y, which is clearly dependent with X, even if y is revealed.  If the training set changes, the learned theta changes.  The reviewer wonders if this can be explained by the theorems in the paper – or did the reviewer miss something (response is welcome here)?

- The above leads to another question: The derivation of the work relies on (4) and (5), but either of which fully expressed the gradient. The impression from there is the updates are dominated by the covariance matrices and if the covariance matrices are identity, then there is no information about the training data passed through the training process. This may not be true, if, again, consider the least squares problem, where the gradient is

     grad = X’*X theta – X’*y

where the second term is still data-dependent. In particular, the claim “To establish this result, we note that the first layer activation at initialization, Z0 train, is a random variable due to random weight initialization, and only depends on Xtrain through Ktrain:” The last sentence is a bit questionable to the reviewer.

- There may be some clarity issues. For example, (8) is not easy to understand. In the test stage, why is there still a gradient step performed? The test stage should not involve any optimization. This may be clarified.

- Basic generalization theorem suggests that generalization is only related to function class’s complexity, but not the correlation among the coordinates of the data samples. As long as the training and test data are sampled from the same distribution, it is hard to see, from classic generalization analysis viewpoint, why using whitening or second-order methods can destroy generalization. The claim from this work is contradicting to what we learn from textbooks, e.g., Shalev-Shwartz, Shai, and Shai Ben-David. Understanding machine learning: From theory to algorithms. Cambridge university press, 2014.

It may be good that the authors compare their results with classic results, e.g., those based on uniform convergence, under some function hypothesis complexity measures (e.g., finite class, VC dimension or even Radamacher complexity). The classic proofs of generalization are insensitive to data whitening or algorithm design, and thus the reviewer feel that there may be a big gap to fill if the claims of this work holds.

- The main results claim that whitening is harmful, the simulation may have suggested otherwise. From most of the figures, the generalization error becomes better and better when the sample size increases. Note that when the sample size increases, the sample correlation matrix converges in probability to the ensemble mean, and the whitening could be performed in a more accurate way. This set of results may suggest that, if whitening is done accurately, then it is fine. When the sample size is small, whitening cannot be done very accurately since sample correlation is not accurately estimated.

The reviewer’s understanding is that the high test error with small sample size is unlikely an effect of whitening, but *inaccurate* whitening, since more noise was brought into the training process. But when the whitening step is performed without too much noise, generalization only depends on function class used and the number of samples, per classic generalization theories. This interpretation seems more consistent with classic generalization theory. The authors may hope to comment on this.


------- after the discussion period ------

I would like to thank the authors for the reply, clarification, and additional experiments. Although I do like the perspectives revealed in this work, many points are still quite unclear to me. For example, the theory seems not be able to explain why whitening does not hurt testing when sample size is large, as demonstrated in the paper.  This work also does not draw connection between sample size and generalization error, which may make the claims a bit incredible. I would encourage the authors to work towards this direction and solidify the contribution.

---

> ### Author Response · Authors · 2020-11-20
> **Response to AnonReviewer2, 2/2**
>
> \> ___It may be good that the authors compare their results with classic results … classic proofs of generalization are insensitive to data whitening or algorithm design.___
>
> A major revelation of modern learning theory has been that classical generalization results, such as those referenced by the reviewer are incapable of explaining the generalization properties observed in practice. This is sometimes referred to as the generalization puzzle [Zhang et. al. ICLR 2017 (arXiv:1611.03530), Neyshabur et. al. NeurIPS 2017 (arXiv: 1706.08947)]. It has been realized that a major shortcoming of classic work is that generalization dynamics of neural networks and even linear models [Belkin et. al. PMLR 2018 (arXiv: 1802.01396)] depends crucially on the optimization procedure (including early stopping) [Neyshabur er. al. (arXiv: 1705.03071)], and structure of the data [Gerace et al. 2020 (arxiv: 2002.09339)], not just the worst case behavior of the function class.
>
> \> ___Note that when the sample size increases, the sample correlation matrix converges in probability to the ensemble mean, and the whitening could be performed in a more accurate way. This set of results may suggest that, if whitening is done accurately, then it is fine. When the sample size is small, whitening cannot be done very accurately since sample correlation is not accurately estimated.___
>
> We emphasize that whitening is **defined** in terms of a finite dataset, rather than the distribution that generated that dataset (eg see https://en.wikipedia.org/wiki/Whitening_transformation ). As such, the whitening we consider in the paper is done exactly and accurately.
>
> The question of what happens if the whitening matrix is allowed to go to its ensemble limit is still interesting, though! To answer this, we repeated our linear model experiment, but using a whitening matrix computed on the entire dataset even when training on a subset of the data. See the new Appendix Figure App 2. We find that using the whole-dataset whitening transform, which we call “distribution whitening”, provides a slight performance improvement, but still performs most similarly to when the whitening transform is computed on only a subset of the data.

---

> > ### Comment · AnonReviewer2 · 2020-11-24
> > **Thanks for the references; do not agree with the first item**
> >
> > 1) A major revelation of modern learning theory .... not just the worst case behavior of the function class.
> >
> > Classical results indeed are pessimistic, since the generalization bounds hold over all distributions. I fully agree with this. On the other hand, classical results, although very pessimistic, shouldn't be wrong. This is still contradicting to the claims in this work. This work's main claim is that "whitening destroys generalization", which seems not to tighten classical results or explaining generalization under more precise prior knowledge, data structure or environment parameters (and thus improving generalization understanding beyond the worst case scenario). Instead, the conclusion here seems to even break the worst case scenario given by classical results. This is something that I do not agree.
> >
> >
> > 2) In any case, the generalization understanding may be more interesting if the authors could relate it to sample size. This seems still elusive in the current version.

---

> > > ### Author Response · Authors · 2020-11-25
> > > **More about generalization**
> > >
> > > Thank you for following up here as well!
> > >
> > > > Classical results, although very pessimistic, shouldn't be wrong. This is still contradicting to the claims in this work … the conclusion here seems to even break the worst case scenario given by classical results. This is something that I do not agree.
> > >
> > > We are confident our results do not contradict classical worst case generalization bounds. (indeed, it would be particularly troubling if they did since our results are also experimentally validated).
> > >
> > > As one example our results satisfy standard Rademacher complexity, function class based bounds as long as the assumptions on the maximal range of weight deviation are met (e.g. see Baratin et. al. https://arxiv.org/abs/2008.00938 for a review of the classical bounds and an attempt to incorporate data structure).
> > >
> > > > the generalization understanding may be more interesting if the authors could relate it to sample size. This seems still elusive in the current version.
> > >
> > > We are interpreting sample size to mean the number of examples in the training set. Let us know if this is not what you intend.
> > >
> > > We derive distinct sets of theoretical results for the small dataset (n < d) and large dataset (n > d) regimes, so we do make some contact theoretically with the role of dataset size. The majority of our experiments additionally probe the role of dataset size by sweeping over it. We completely agree, though, that a more nuanced theoretical analysis of the role of sample size is an important and interesting topic for future work.
> > >
> > > As always, if we can provide any additional information or clarification (before the end of the discussion period later tonight), please let us know. Thank you for taking the time for a careful review!

---

> ### Author Response · Authors · 2020-11-20
> **Response to AnonReviewer2, 1/2**
>
> Thank you for your careful review, and especially for the specific actionable requests for clarification. We believe we have addressed your concerns below, and hope you consider raising your score as a result. If we have not fully addressed your concerns, please follow up!
>
> \> ___The reviewer considered a simple least squares problem … the solution is theta_opt = X_transpose\*y, which is clearly dependent with X, even if y is revealed. ... The reviewer wonders if this can be explained by the theorems in the paper___
>
> We thank the reviewer for their question about linear least squares. Up to a transpose, and a use of W rather than theta, this example is precisely the linear model discussed in Section 2.4 and Appendix B. It is not a contradiction of the main message of the paper.
>
> For notational consistency with the paper, we will consider the linear model f = W X, and temporarily assume that the input dimensionality is smaller than the dataset size, d < n. The optimal parameters are given by W_{opt} = y_{train} X_{train}^+,
> where the superscript + indicates the pseudoinverse. (Note that if X_{train} is whitened, we have X_{train}^+ = X_{train}^T, and this agrees with the expression you write in your review.) This captures the dependence of W_{opt} on X_{train}. However, though not immediately obvious, the dependence of the function f(X; W_{opt}) on X_{train} is much more limited! As a brief illustration we compute the function outputs f_{test} for test inputs X_{test}:
> f_{test} = W_opt X_{test}
>               = y X_{train}^+ X_{test}
>               = y X_{train}^+ X_{train}^{T+} X_{train}^T X_{test}
>               = y (X_{train}^T X_{train})^+ X_{train}^T X_{test}
>               = y K_{train}^{+} K_{train x test}.
>
> So in this case, the test set predictions depend only on the second moment matrices K_{train} and K_{train x test}, consistent with the theorems in the paper.
>
> To clarify this example further in the text, we have included expanded discussion of the properties of the global optimum $w^{\star}$ in Appendix B.
>
> \> ___The derivation of the work relies on (4) and (5), but either of which fully expressed the gradient.___
>
> We’re not sure if we understand your concern in this case. It may be addressed by noting that Equations 4 and 5 are written for general loss functions (they don’t rely on an MSE loss), and the arguments we present in Sections 2.1-3 are independent of the specific form of the loss. If that doesn’t address your question, please ask for further clarification!
>
> \> ___This may not be true … consider the least squares problem, where the gradient is
> grad = X’X theta – X’y___
>
> This gradient leads to the update step for W^t in Equation 4, in the case that L is a square loss. You are correct that after training the first layer weights W contain information about X which is not included in K. The key is that *the activations Z = W X depend only on K*. That is, the first layer weights W depend on X in precisely the right way such that the following activations Z, the remainder of the parameters theta in the network, and the function output f() only depend on X through K. See Equations 5 and 8 in the text.
>
> \> ___“... Z0 train, is a random variable due to random weight initialization, and only depends on Xtrain through Ktrain:” The last sentence is a bit questionable to the reviewer.___
>
> This result corresponds to Equation 7 in the text, and is proved rigorously in Appendix A.
>
> \> ___[Equation] (8) is not easy to understand. In the test stage, why is there still a gradient step performed?___
>
> In Equation 8, we are looking at the time evolution of the test set predictions over the course of training. Training itself is carried out on the training set. We use Equation 8 to identify sources of data dependence in the test predictions over the course of training. We have updated the text to clarify this.

---

> > ### Comment · AnonReviewer2 · 2020-11-24
> > **Thanks for the clarification; still some confusions left**
> >
> > I'd like to thank the authors for their patience and right-to-the-point response. I'm still confused by a some points.
> >
> > 1) f_{test} = ... = y K_{train}^{+} K_{train x test}.
> >
> > The derivation should be correct. But K_train x test is basically X_train^T X_test, which depends on X_train, not contradicting to my previous comment. This still does not quite clearly explain why whitening "destroys" generalization. In addition, test is conducted at a sample by sample sense. Not sure if K_{train x test} is insight revealing.
> >
> > It is also a bit unclear to the reviewer if X_test is whitened or not. Is it better to express X_train_raw as unwhitened data. and X_train = X_train_raw U, where U is the whitening matrix. Similar X_test = X_test_raw x U (not sure if they use the same U)? If this is the case, K_{train x test}=U^T X_train_raw X_test_raw U. Does this U bring harm to generalization?
> >
> > 2) The key is that the activations Z = W X depend only on K. .... see Equations 5 and 8 in the text.
> >
> > Thanks for the patience for explaining. I would like to always think about the least squares case. Consider
> > min ||Z-WX||^2. Then, the activation, in the least squares cause, should reside in the orthogonal complement of range(X_{train}^T). From this viewpoint, it does not matter if you have whitened X_train, since the whitening does not change the range space.

---

> > > ### Author Response · Authors · 2020-11-25
> > > **Additional clarifications**
> > >
> > > Thank you very much for following up with us!
> > >
> > > > It is also a bit unclear to the reviewer if X_test is whitened or not. Is it better to express X_train_raw as unwhitened data. and X_train = X_train_raw U, where U is the whitening matrix. Similar X_test = X_test_raw x U (not sure if they use the same U)? [rest of comment answered below]
> > >
> > > Yes, the same whitening transform $U$ is applied to train and test (and validation) data. The difference between our conditions is just whether $U$ is computed using training data only, or using all the data. We will edit the paper text to emphasize this.
> > >
> > > A nit which could cause confusion below: our convention is that $X$ has shape [# features] x [# samples], so the above relationships should be $X = U X_{raw}$, with $U$ multiplied on the left.
> > >
> > > > K_train x test is basically X_train^T X_test, ... This still does not quite clearly explain why whitening "destroys" generalization. In addition, test is conducted at a sample by sample sense. Not sure if K_{train x test} is insight revealing.
> > >
> > > I think this example is an excellent testbed. For simplicity of discussion, let’s consider the full data whitening condition (first subsection of section 2.3). After $X$ is whitened $K$ becomes the identity matrix, that is
> > > $K_{ij} = 1$ if $i=j$, and $0$ otherwise,
> > > where $K$ is the full second moment matrix, containing both training and test data. $K_{train\ x\ test}$ is an off diagonal block of $K$ though -- so every entry in $K_{train\ x\ test}$ will be 0. Because $K_{train\ x\ test}$ is a zero matrix, $f_{test}$ will predict zero for all test points when full data whitening is performed.
> > >
> > > Because this is such a good example, we will update our paper to include explicit discussion of it (this may not happen until after the end-of-discussion deadline tonight).
> > >
> > > > … If this is the case, K_{train x test}=U^T X_train_raw X_test_raw U. Does this U bring harm to generalization?
> > >
> > > See nit from above: this should be K_{train x test}=X_train_raw^T U^T U X_test_raw
> > >
> > > It is possible to express the results in our paper in terms of the action of U as you suggest. I believe this would be awkward and complex though, so I will not try to do it here. Instead, I think the core important point is that $f()$ only depends on $X$ through $K$. That is, the function computed by the network is really $f(K(X))$, and if $K$ is made trivial or uninformative, then the function $f()$ computed by the network must also be uninformative. $K$ is more trivial for whitened data than for unwhitened data, and so $f()$ will tend to be also.
> > >
> > > > Thanks for the patience for explaining. I would like to always think about the least squares case. Consider min ||Z-WX||^2. Then, the activation, in the least squares cause, should reside in the orthogonal complement of range(X_{train}^T). From this viewpoint, it does not matter if you have whitened X_train, since the whitening does not change the range space.
> > >
> > > Yes! Good observation. It is true that because optimization occurs in the same subspace the solution reached if the loss is minimized is the same for whitened and unwhitened data.
> > >
> > > The key is that the *trajectory* taken to reach the global minimum is different when training is done on whitened or unwhitened data. So if training is performed with early stopping based on validation error (as is common in machine learning), then the model trained on unwhitened data typically generalizes better.
> > >
> > > Check out Figure 4a. We train a linear model by gradient descent on whitened and unwhitened data. If you train for long enough, both models reach the same solution and achieve the same (poor) performance. If you stop training before the models have fully converged though, then the test error is better for the model trained on the unwhitened data than the one trained on the whitened data.
> > >
> > > Thank you again.

---

### Official Review · AnonReviewer1 · 2020-10-29
**Under *some* conditions, whitening and second order methods *may* not generalize well**

**Rating:** 4
**Confidence:** 5

**Review:**

This paper shows theoretical and empirical evidence that under some conditions, whitening the input data and second-order methods may hurt the generalization performance. The paper is well written with good intuitions but overclaims its results. The theoretical results are limited to specific cases, and the experimental results would benefit from systematic ablation studies. All in all, I do not think that the paper provides sufficient evidence that justifies its title and main message of the paper. Detailed review below.

- In Section 2.1, "the trained model depends on the training data only through K", this is technically incorrect, since there is a dependence on y_train and the Z_train is independent of X_train, only when conditioned on both K_train and y_train. Since whitening only affects K_train, there still might be some information in y_train, especially for the case of data realizable by the model. Please clarify this.

- In Section 2.3, the statement "whitening hurts generalization" is true only for high-dimensional datasets, where the input dimension > number of samples. I agree that such datasets (arising for example in genetics) are important, this is not the typical case where neural networks are used. For example, for all vision tasks, d < n. Please explicitly say this.

- In Section 2.4, please also consider d > n case. This is more important because there are multiple solutions, even in the linear case, and whitening may hurt generalization. For the d < n case analyzed in the paper, for a linear model, for most losses, the optimization problem is strictly convex and a unique solution. This is hardly a claim justifying that "whitening hurts generalization", it is an optimization problem and is orthogonal to the paper. For convex problems with a unique solution, there is no reason to early stop the optimization. The convergence rate is linear implying that the solution is reached quickly and this result is not interesting, either from an optimization/generalization perspective. Moreover, such a claim is only valid for the squared loss, and not, for example, the logistic loss. Please clearly explain this.

- In Section 2.5, the paper claims that "unregularized second-order methods have poor generalization", but as the authors admit themselves, second-order methods are rarely used without regularization, again making this result not practically important. Moreover, there have been second-order methods (KFAC for example) that employ regularization, approximations to the Hessian and generalize well. The statement "second-order information destroys generalization" is therefore an overclaim.

- Moreover, this section focuses on the squared loss with a linear model. If n > d, then Newton's method converges to the min-norm solution, same as GD. For d > n, the solution of Newton's method lies in the span of the training data (if we use the pseudo-inverse for the Hessian or add a small regularizer) and also converges to the min-norm solution, meaning in both cases, its generalization is as good as GD. Again, the behaviour for other losses is not clear.

- In Section 3, for the experimental results, there are numerous confounding factors that need to be controlled for. For example, is the optimization deterministic or stochastic? We know that stochastic methods typically generalize better. How is the learning rate selected? Is there a warmup phase? These factors play an important role in the generalization. Similarly, both the optimization method and the loss function also influence the generalization. Is there regularization, either L2 or L1 in the case when d > n (this is typically done for high dimensional datasets). Please explain how did you control for these factors.

- In Figure 3(a), what is the effect of using a regularized Newton method? What if you run a more standard second-order method like KFAC?

- In Figure 5, the paper shows "Regularized second-order methods can train faster than gradient descent, with minimal or
even positive impact on generalization". This contradicts the paper's title. As the authors claim, when done without regularization, second-order methods can harm generalization, but when used with proper regularization, second order methods help. Like I said before, there are a number of confounding factors and the story is not as simple.

---

> ### Author Response · Authors · 2020-11-20
> **Response to AnonReviewer1, 2/2**
>
> \> ___In Section 2.4 … such a claim is only valid for the squared loss, and not, for example, the logistic loss. Please clearly explain this. … In Section 2.5 … this section focuses on the squared loss with a linear model … the behaviour for other losses is not clear.___
>
> Thank you for catching this! We have clarified that the discussion in Section 2.4 is valid for the squared loss (we had accidentally moved the key sentences into Appendix B).
>
> In Section 2.5 we have also expanded our discussion around our assumption of a squared loss, to: further emphasize that the theory results only hold for squared loss; but also to foreshadow that we also observe the effect experimentally for cross entropy loss.
>
> For casual readers (we recognize the reviewer already understands this) -- we would also like to emphasize that our theory results in Sections 2.1, 2.2, and 2.3 hold for any loss.
>
> \> ___Newton's method converges to the min-norm solution … meaning in both cases, its generalization is as good as GD.___
>
> This is incorrect. Similar to our above discussion, GD on unwhitened data can outperform Newton’s method, or GD on whitened data, precisely because the early stopped performance of GD is better. This is not the case for Newton’s method. The fact that GD on whitened and unwhitened data and Newton’s method find the same optimum if run to convergence is true, but not key to why linear models trained on unwhitened data by GD, if early stopped, perform better.
>
> \> ___In Section 3, for the experimental results, there are numerous confounding factors that need to be controlled for… Please explain how did you control for these factors.___
>
> All the details of our experiments are given in Appendix E. The information in this appendix addresses your specific concerns such as the method by which the learning rate was chosen, whether optimization was stochastic or deterministic for each experiment, etc. If you have further questions please follow up.
>
> \> ___In Figure 5, the paper shows "Regularized second-order methods can train faster than gradient descent, with minimal or even positive impact on generalization". This contradicts the paper's title.___
>
> There is no contradiction. Second order methods destroy information that could otherwise be used for generalization. Whether this destruction of information that could be used for generalization is harmful in practice depends on the specifics of the model, task, and optimizer. In addition to Figure 5, we also discuss this in the abstract and in Section 3.
>
>
> Thank you again for your time and detailed review.

---

> ### Author Response · Authors · 2020-11-20
> **Response to AnonReviewer1, 1/2**
>
> Thank you very much for your time and effort! We believe we have addressed all of your technical concerns, and hope you consider raising your score as a result. If we have not successfully addressed your concerns, please ask for further clarification!
>
> \> ___In Section 2.1, "the trained model depends on the training data only through K"___
>
> Thank you for catching this! We have modified the text to state “the trained model depends on training inputs only through K”.
>
> \> ___In Section 2.3, the statement "whitening hurts generalization" is true only for high-dimensional datasets, where the input dimension > number of samples.___
>
> Thank you for your suggestion. We have changed the subsection title to “Whitening harms generalization in high dimensional datasets” to match the theory results in the section. We also wish to emphasize that we show theoretically in Section 2.4 that whitening can harm generalization in linear models when [input dimension] < [# samples], and experimentally in Figure 3 that whitening can harm generalization in nonlinear models when [input dimension] < [# samples].
>
> We have also made the textual change suggested for Section 2.3 to note that many neural net applications do not involve high dimensional datasets relative to the number of training examples. We placed this change in the Introduction.
>
> \> ___This is hardly a claim justifying that "whitening hurts generalization", it is an optimization problem and is orthogonal to the paper. For convex problems with a unique solution, there is no reason to early stop the optimization.___
>
> This is incorrect. Early stopping even for convex problems can mitigate over-fitting and lead to significant performance gains (see Figure 4a for example). Early stopping is commonly used in optimization (eg. as described in Chapter 8 in Goodfellow et al, “Deep Learning”, 2016). One way to phrase the main message of Section 2.4 is that for unwhitened data one can leverage the performance gains from early stopping while for whitened data one cannot. Indeed, at the global optimum, there is no performance difference between whitened and unwhitened data.
>
> \> ___The convergence rate is linear implying that the solution is reached quickly and this result is not interesting.___
>
> Differences in linear convergence rates are practically interesting. For linear models, the convergence rate of gradient descent is inversely proportional to the condition number of the Hessian. The Hessian condition number is often exponentially large in the input dimensionality. Whitening and second order methods can thus dramatically accelerate training, since they can converge linearly at a rate that is often many orders of magnitude better than steepest descent.
>
> \> ___The statement "second-order information destroys generalization" is therefore an overclaim.___
>
> This is a mis-quote. We never say "second-order information destroys generalization". We say that whitening and second order optimization destroy *information* about the dataset. This is true even when input dimensionality is less than dataset size. Information in the feature-feature second moment matrix is made unusable and inaccessible to the training procedure. Whether destroying information which could be used for generalization harms generalization in practice depends on specifics of architecture and training procedure, as we discuss in the paper.
>
> \> ___please also consider d > n case. This is more important because there are multiple solutions, even in the linear case, and whitening may hurt generalization.___
>
> We have now included a discussion of the case d > n case in Appendix B. As noted by the reviewer, though there are formally multiple solutions to the optimization problem, there is a unique solution found by GD, SGD, and projected Newton’s method and it is the same for whitened and unwhitened data. In this case, as in the n <= d case, the benefit of unwhitened data comes entirely from early stopping and not from properties of the optimum. We hope this is more clear in our expanded discussion.

---

### Decision · Program_Chairs · 2021-01-07
**Final Decision**

**Decision:**

Reject

**Comment:**

The paper suggests that whitening the data harms generalization and optimization performance when learning models of the form h(W x) i.e. those that are based on a linear projection of the inputs (which includes DNNs for instance). The main concern of the reviewers is that their theoretical developments were not that convincing; it seemed more along the lines of providing some specific anecdotes. But more broadly, the caveat is that their development seems very simple: their results in high-dimensional settings (where d = dim(x) > number of samples n) is not that relevant since vanilla whitening is anyway fraught in such high-dimensions, since the sample covariance matrix is not a good estimator in high dimensions anyway. And when n > d, their result focuses on linear models, where they say that whitening reduces information about the singular vector directions where the input data might mostly lie on. But if the data lies on a lower dimensional linear manifold, then whitening is again fraught: the covariance matrix is singular. The linear manifold assumption also seems very specific given the general title of the paper. Overall, the paper needs to narrow their focus on specific settings where whitening is harmful, but the specific settings above in and of themselves do not necessarily say anything other than to estimate the covariance matrix carefully before doing whitening.